# The importance of input data quality and quantity in climate field reconstructions – results from the assimilation of various tree-ring collections

Franke, Jörg[1,2], Valler, Veronika[1,2], Brönnimann, Stefan[1,2], Neukom, Raphael[1,2,3,4], Jaume Santero, Fernando[5]

[1] Institute of Geography, University of Bern, Switzerland.

[2] Oeschger Centre for Climate Change Research, University of Bern, Switzerland.

[3] Department of Geography, University of Zurich, Switzerland.

[4] Department of Geosciences, University of Fribourg, Switzerland

[5] Universidad Complutense de Madrid.

*Correspondence to: Jörg Franke (franke@giub.unibe.ch)*

**Abstract.** Differences between paleoclimatic reconstructions are caused by two factors, the method and the input data. While many studies compare methods, we will focus in this study on the consequences of the input data choice in a state-of-the-art Kalman-filter paleo data assimilation approach. We evaluate reconstruction quality in the 20[th] century based on three collections of tree-ring records: (1) 54 of the best temperature sensitive tree-ring chronologies chosen by experts; (2) 415 temperature sensitive tree-ring records chosen less strictly by regional working groups and statistical screening; (3) 2287 tree-ring series that are not screened for climate sensitivity. The three data sets cover the range from small sample size, small spatial coverage and strict screening for temperature sensitivity to large sample size and spatial coverage but no screening. Additionally, we explore a combination of these data sets plus screening methods to improve the reconstruction quality.

A large, unscreened collection leads generally to a poor reconstruction skill. A small expert selection of extratropical northern hemisphere records allows for a skillful high latitude temperature reconstruction but cannot be expected to provide information for other regions and other variables. We achieve the best reconstruction skill across all variables and regions by combing all available input data but rejecting records with insignificant climatic information (p-value of regression model > 0.05) and removing duplicate records. It is important to use a tree-ring proxy system model that includes both major growth limitations, temperature and moisture.

## 1 Introduction

In the past 20 years, a lot of effort has been invested in improving climate reconstructions for the last centuries to millennia based on indirect climate information – so-called "proxies". Focus has been on both, large-scale averages as well as the reconstructions of regional to global fields (Masson-Delmotte et al., 2013; Smerdon and Pollack, 2016). Temporal and spatial resolution varied with the included paleoclimatic archives. However, most reconstructions for the past centuries rely heavily on the most abundant indirect climate archive, tree rings, and specifically on tree-ring width (TRW) and late-wood density (MXD). Differences between reconstructions have

mostly been discussed with differences in reconstruction methodology in mind (Christiansen and Ljungqvist, 2017). However, a new study shows good agreement between a wide range of methods, if reconstructions are based on the same input data set (Neukom et al., 2019a; 2019b). Another recent study found that temperature sensitive tree-ring proxies from the PAGES2k database (Emile-Geay et al., 2017) lack multi-centennial trends, which are found in other proxy archives (Klippel et al., 2019). This suggests that the input data play a crucial role for differences between reconstructions. This fact is also seen in data assimilation for weather prediction, e.g. at the addition of satellite to radiosonde observations (Swinbank et al., 2012, p.365). Today, many proxy data archives are available, hence compiling input data for reconstruction is not only a matter of the amount of proxy data, but also of their selection, i.e., screening.

In this study, we therefore aim at evaluating the effect of various tree-ring data collections and their screening on the final reconstructions. Tree-ring proxies are by far the most numerous climate information source for the past centuries and additionally chosen because our methodology relies on annual data without dating uncertainties. Due to the relevance of temperature in the climate change discussion and the fact that many biological proxies react to temperature stress, temperature has so far been the variable of most interest. However, to study the underlying processes a multi-variable perspective is required. Therefore, we evaluate the effects of the input data choice, using a state-of-the-art data assimilation technique, which allows for multi-variable climate reconstructions in form of model simulations that are in optimal agreement with proxy information (Bhend et al., 2012; Franke et al., 2017).

A number of previous studies based on data assimilation techniques tended to assimilate a high quantity of input data instead of applying a strict data selection beforehand (e.g. Steiger et al., 2018; Tardif et al., 2019). The idea is that regression-based proxy system models weight each proxy series by their regression residuals. Hence, proxies that do not contribute information will be downweighted automatically. However, this weighting may not work perfectly because of two factors: (1) the regression depends on overlapping paleodata and instrumental measurements, which often results in a small sample (Fig. 1 in Jones et al., 2012), uncertain residuals and possible model overfitting; (2) moisture and temperature sensitive proxies may correlate and hence moisture sensitive paleodata will be used to correct temperature and vice versa. However, these two variables probably have very different multi-decadal to centennial variability (Franke et al., 2013). The growth limiting factor may even change over time (Babst et al., 2019).

In this study, we use the Kalman filter based state-of-the-art data assimilation technique introduced in Bhend et al. (2012), which is very similar to the methodology used in the last millennium reanalysis (LMR) project (Hakim et al., 2016; Tardif et al., 2019). In contrast to LMR, our method is a transient-offline method, in which the background state is time-dependent due to the external forcing prescribed to the climate model simulations. In our experiments, we focus on the effect of the input data choice on the final reconstruction. We compare three published collections of tree-ring records (focusing on TRW and MXD), of which at least two are commonly used for climate reconstructions. These have very different characteristics: (1) The B14 collection of 2287 consistently detrended TRW chronologies from the International Tree Ring Data Base (ITRDB), not screened for climate sensitivity (Breitenmoser et al., 2014); (2) TRW and MXD series from the PAGES2K database version 2 (Emile-Geay et al., 2017), with a selection of 415 temperature sensitive records, most selected by a statistical screening for positive correlation with instrumental temperature; and (3) the N-TREND tree-ring collection of 54 TRW,

MXD or blended TRW-MXD time series (Wilson et al., 2016), selected by experts to be the best temperature
recorders. Thus, the three data sets cover the range from large sample size and spatial coverage but no screening
for temperature sensitivity to small sample size and small spatial coverage but strict screening. Note, that these
collections were generated with slightly different aims, which affects their use in reconstructions. Thus, for
instance, we cannot expect to achieve the best global scale field reconstruction already from a proxy collection
covering a much smaller area (Kutzbach and Guetter, 1980). However, all data sets are used for climate
reconstruction.

In the next section the method and data sets are introduced in greater detail before we show our results. Then we
discuss the possible reasons for our results and the differences compared to previous studies. Finally, we draw our
conclusion how an optimal proxy selection process should look like.

## 2 Data and Methods

We use three input data sets for comparison, all consist of annually resolved tree-ring measurements, which have
hardly any dating uncertainties:

1.  B14 is a collection by Breitenmoser et al. (2014) of 2287 uniformly detrended and standardized TRW
    measurements from the ITRDB (Zhao et al., 2018). We use the full collection without any further
    screening for climate/temperature sensitivity. Hence, this represents the data set with the highest quantity
    of records. However, the weighting of temperature information in the paleodata is completely up to the
reconstruction method.
2.  PAGES2k is a collection of 415 TRW and MXD series from PAGES2k data base version 2 (Emile-Geay
    et al., 2017). These are all records that correlate significantly (p<0.05) with nearby instrumental
    temperature measurements and/or have been described by experts to represent temperature variability.
    This compilation represents a compromise of good quantity, large spatial coverage and good quality
paleodata, based on global selection criteria. However, experts from various regional groups were
    differently strict in their screening procedure, which led to varying data density in the different regions.
3.  N-TREND is a collection of 54 tree-ring chronologies based on TRW, MXD or a combination of both.
    They were chosen by experts to be just the best tree-ring paleodata for temperature reconstructions
    (Wilson et al., 2016). Hence, they are our low quantity, highest quality input data set with least spatial
coverage.

Climate fields are reconstructed by assimilating these tree-ring observations into an ensemble of climate model
simulations using a Kalman filter technique, Ensemble Kalman Fitting (Bhend et al., 2012; Franke et al., 2017).
The simulations, which serve as a background (sometimes called first guess or prior) of the atmospheric state at
each point in time, are given by a 30-member initial condition ensemble of atmospheric model simulations
(ECHAM5.4, (Roeckner, 2003)). All simulations follow the same external forcings (volcanic (Crowley et al.,
2008), solar (Lean, 2000), greenhouse gases (Yoshimori et al., 2010), land use (Pongratz et al., 2008), tropospheric
aerosols (Koch et al., 1999)) and sea surface temperatures boundary conditions based on a reconstruction by(Mann
et al., 2009) plus additional El Niño Southern Oscillation variability (Franke et al., 2017). The data assimilation
method is "transient offline". "Transient" refers to the fact that our prior at each point in time consists of 30
ensemble members that are in agreement with forcings and boundary conditions. "Offline" assimilation means

that the simulations are calculated for the full period before the assimilation is conducted. This is possible in the paleo-climatological setup because we have only one observation per year per record. Predictability on these time scales only comes from the boundary conditions and not from the atmospheric model.

EKF is the offline variant of the ensemble square root filter (Whitaker and Hamill, 2002) in which the observations ($y$) are assimilated serially. The assimilation procedure is divided into an update of the ensemble mean ($\bar{x}$) and an update of the anomalies with respect to the ensemble mean ($x'$):

(1) $\quad \bar{x}^a = \bar{x}^b + K(\bar{y} - H\bar{x}^b)$

(2) $\quad x'^a = x'^b + \widetilde{K}(y' - Hx'^b) = (I - \widetilde{K}H)x'^b \ with: y' = 0$

where the superscript [a] refers to the analysis and [b] to the background of the atmospheric state x, which is a vector with values of multiple variables at all grid boxes. $H$ denotes an operator which maps $x^b$ to the observation space (see proxy system model PSM below). $K$ and $\widetilde{K}$ are the Kalman gain matrices (Whitaker and Hamill, 2002):

(3) $\quad K = P^b H^T (HP^b H^T + R)^{-1}$

(4) $\quad \widetilde{K} = P^b H^T \left[\left(\sqrt{HP^b H^T + R}\right)^{-1}\right]^T \times \left(HP^b H^T + R + \sqrt{R}\right)^{-1}$

The K matrices control how the information from the observations updates the background. It depends on the observation error covariance matrix $R$ and the background error covariance matrix $P^b$. $R$ is estimated from the regression residuals of the PSM and errors are assumed to be uncorrelated. Background error covariances $P^b$ are calculated from the 30-member ensemble $n_{ens}$ of ECHAM5.4 simulations (CCC400) at each time step with row number $i$, column number $j$ and ensemble member $k$ (Bhend et al. 2012, Franke et al. 2017):

(5) $\quad P_{i,j}^b = \frac{1}{n_{ens}-1} \sum_{k=1}^{n_{ens}} x_{i,k}'^b x_{j,k}'^b$

This has the advantage of taking time-dependent covariance structures into account, for instance during El Niño vs La Nina years. The disadvantage is the small sample of 30 ensemble member for covariance estimation. To deal with spurious correlations caused by the relatively small ensemble, we apply a distance-dependent localization, i.e. updates are only possible with a certain radius around the observations (Valler et al., 2019):

(6) $\quad C = exp\left(-\frac{|d_i - d_j|^2}{2L^2}\right)$

$d_i$ and $d_j$ describe the zonal and meridional distances from the selected grid box. $L$ is the length scale parameter used for localization. It has been estimated based on the spatial correlation in the simulations and is variable dependent, e.g. 1500/450 km in case of temperature/precipitation (Franke et al., 2017).

A recent comparison of Valler et al. (2018) has shown superior performance when using an improved covariance estimation, which blends 50% of the 30-member time-dependent covariance with 50% of a 250-member "climatological" time-independent covariance (Experiment: 50c_PbL_Pc2L in Valler et al. (2018)). In this paper we use both the original setting as in Franke et al. (2017) as well as the improved setting proposed by Valler et al. (2018).

Our paleo-reanalysis is based on anomalies from a 71-year period around the current year. Low frequency

variability is a function of the models' response to the prescribed external forcings and background conditions, which include sea-surface temperatures. Because low frequency variability is not consistently preserved in paleodata (Franke et al., 2013; Klippel et al., 2019), but reasonably well represented in the model simulations of the last millennium (Franke et al., 2017), this approach is expected to provide consistent skill at all time scales. Note that the assimilation of anomalies retains possible model biases. This circumvents a big problem in data assimilation approaches with temporally varying input data networks. Observations that gradually pull the model away from its biased state, can lead to artificial trends or step functions in time-series.

We use a linear multiple regression PSM to simulate tree-ring observations using modelled temperature and/or precipitation. The regression model is calibrated with gridded instrumental data (CRU TS 3.1, Harris et al., 2014) in the period 1901-1970. It includes monthly temperature (and precipitation) during the growing season April to September. In this study, we limit the analysis to the northern hemisphere because the majority of the tree-ring observations can be found there. In the first four experiments (Table 1), which only use temperature (T) in the PSM, we have 6 independent variables (i.e., local, monthly mean temperature of April to September). If we assume that tree growth was limited by temperature and moisture (TR) variability (experiments 5 and 6 in Table 1), we have 12 independent variables (i.e., local, monthly mean temperature of April to September, and monthly precipitation sums of April to September). Note that regression coefficients can be zero and thus growth can still be limited to just temperature or just precipitation and to less than 6 months. In experiments 7 and 8 (Table 1) we additionally consider only regression models, in which the growth occurs in consecutive months. Therefore, we fit all possible combinations of consecutive months and choose the PSM with the lowest Akaike information criterion (AIC). Temperature and precipitation limitations can occur in a different sequence of months for each variable (e.g. precipitation limits growth from April to June and temperature limits growth from June to September). The variance of the regression residuals is used to specify the observation error covariance matrix (assumed diagonal) in the assimilation, i.e. the larger the residuals, the less weight an observation gets and the less the model simulations get corrected.

In this study, the period in which the regression coefficients of the PSM are estimated as well as the calculations of the regression residuals overlaps with the period when the reconstruction skill is estimated. This apparent lack of independence is negligible in this case because: regression coefficients are estimated from gridded instrumental data sets to translate grid cell temperature (and moisture) anomalies to local tree-ring measurements. The optimization is done on tree rings, not on the climate data, and it is done on many local scales. In that sense the effects of the dependence are rather indirect. In contrast to statistical reconstruction methods, which directly estimate a climate variable such as temperature through the regression parameter estimate, our assimilation method is far less affected by the calibration procedure. Nevertheless, using the same data for validation probably lead to a slight overestimation in reconstruction skill. However, in this study we just compare the relative skill of various inputs data sets, so the impact of dependencies will be the same for all. Concerning the regression residuals, again the error estimate concerns tree ring width, not climate parameters. We use the residuals as an estimate of error covariance. In case we underestimate the residuals, proxy observation would have a too much weight in the assimilation process compared to the simulations. Uncertainty estimation in both, observations and models, is a crucial but challenging part of data assimilation. We evaluate the spread-to-error ratios to assess the under/overconfidence of our reconstructions (Franke et al., 2017).

If multiple data collections are combined, there may be duplicates of the same proxy, possibly in differently treated/detrended versions. We conduct experiments where we prevent single sites from being assimilated twice by only choosing the best proxy (smallest regression residuals irrespective of series length) in a 0.1°x0.1° (ca. 10km) grid. This is a rare case, however, and hardly effecting the results.

We evaluate the quality of the reconstruction based on correlation with gridded instrumental observations of temperature, precipitation (Harris et al., 2014) and sea level pressure (Allan and Ansell, 2006) in the period 1901-1990 as a reference ($x^{ref}$, where $x$ is the state vector). After showing absolute correlation coefficients of the analysis, we focus on correlation improvements over the original model simulations, because these forced simulations already correlate positively with the gridded observations in many locations. Correlation focuses on the co-variability, i.e. the correct sign of the anomaly. Additionally, we use a root-mean-square-error skill score $RMSESS$ that describes the improvement of the analysis $x^a$ over the original model simulations (background) $x^b$ over all time steps $i$:

$$(7) \qquad RMSESS = 1 - \frac{\Sigma\left(x_i^a - x_i^{ref}\right)^2}{\Sigma\left(x_i^b - x_i^{ref}\right)^2}$$

It is more difficult to reach positive $RMSESS$ values than correlation improvements, because this score penalizes a wrong amplitude of variability (e.g., an uncorrelated reconstruction with correct variance would yield $RMSESS$ = -1). Because it is based on squared errors, too high variability is penalized more than little variability, which the ensemble mean of the original model simulations has. We only present correlation improvements and $RMSESS$ of the ensemble mean. In contrast to correlation coefficients, which tend to be higher for the ensemble mean than for the ensemble members, $RMSESS$ of single ensemble members tends to be slightly higher than $RMSESS$ of the ensemble mean (Fig. 6 in Bhend et al., 2012).

To evaluate the influence of the input data on the final reconstruction, we conducted the following set of experiments:

Table 1: Experiments

| | Name | Proxy system model | Description |
|---|---|---|---|
| 1. | NTREND_T | 6 regression coeff. for Apr. to Sep monthly temperature (T) | Using the best tree-ring chronologies for temperature reconstruction, which have been chosen by experts, i.e. very strict selection of few, best records |
| 2. | PAGES_T | Same as above | Using a selection of temperature sensitive proxies selected by the regional PAGES working groups. The mostly statistical screening for a temperature signal involved a sign correction, i.e. if temperature and moisture are negatively correlated, tree-ring chronologies can remain as temperature sensitive in the data set. Therefore, more records but less strictly screened than NTREND. |
| 3. | B14_T | Same as above | Consistently detrended tree-ring data from the ITRDB by B14. This proxy set includes the largest amount of proxy series. However, many of them do not include any climate signal. |
| 4. | ALL_T | Same as above | All three data sets together, largest data set with greatest spatial coverage. However, duplicate proxies have not be excluded |

| 5. | ALL_TR | 12 regression coeff. For Apr. to Sep. monthly temperature and precipitation (R) | Same as above |
|---|---|---|---|
| 6. | ALL_TR_scr0.05 | Same as above | Same as above but with additional basic statistical screening, i.e. only records with a climate signal (p-value < 0.05) will be assimilated. This procedure removes records without or with very little and uncertain climatic information. |
| 7. | ALL_TR_scr0.05_AIC_NOdup | Maximum of 12 regression coefficients but only consecutive months are allowed, still mixed temperature and precipitation signals are possible | Same as above but we chose with the AIC the regression model under the precondition that only climate from consecutive months can influence tree growth, which is more realistic due to local growing season length. Additionally, we remove duplicate proxies by only considering the best proxy (lowest regression residuals) within a 0.1°x0.1° (ca. 10 km) grid. In each grid box we keep both, the best mainly temperature limited and the best mainly moisture sensitive proxy if both exist. |
| 8. | ALL_TR_scr0.05_AIC_NOdup_ClimCovar | Same as above | Same as above but with background error-covariance estimate not only from the 30 ensemble members of the current year. Instead we use a mix of 50% error covariance coming from 250 random ensemble members and years. |

## 3 Results

Temperature correlation coefficients between the analyses and gridded instrumental data are positive nearly all around the globe and for all three proxy collections (Fig. 2a,b,c) because the transient simulations follow forcings and boundary conditions and hence show proper multidecadal variability and a 20th century warming trend. However, this is not the case for precipitation, which does not show a warming trend (Fig. 2d,e,f). In contrast to the assimilation of PAGES and NTREND (Fig. 2d,e), we can observe clearly higher correlations in the United States if the B14 proxies are assimilated (Fig. 2f). Although these first three experiments only use a temperature PSM, information can spread to other variables through the covariance matrix.

To evaluate the differences between the experiments due to the data assimilation we focus on correlation improvement over the background (i.e. the model simulations, which already correlate with the reference data set mainly due to the specified SSTs and external forcing). First, we compare the role of the choice of the three input data sets assuming only temperature dependence and no constraint on the regression model structure (Fig. 3a,b,c; experiments NTREND_T, PAGES_T and B14_T). The highest local improvements are reached with the NTREND data set, however the largest spatial coverage of improvement is found with the B14 data set. Note that temperature correlation improves with all data sets and decreases nowhere, although some proxy records in the B14 data set do not contain any temperature signal. This has been identified with negative regression coefficients for the majority of B14 tree-ring series in the United States of America. In terms of correlation the data assimilation scheme appears to weight the input data appropriately. Looking at precipitation and sea-level pressure correlation improvements (Fig. 4 and 5a,b,c), we find hardly any improvements with the NTREND collection. In contrast, the B14 data set leads to some precipitation correlation improvements over North America, where no

NTREND series are located. Sea-level pressure correlations improve in some regions such as Europe but decrease in other regions like most of Asia (Fig. 5c).

The correct sign of the anomaly, measured by correlation, is only telling us one aspect of the reconstruction quality. To see if the amplitude of the anomaly is also reconstructed correctly, we look at the *RMSESS* skill score (see methods). Here, we find large differences between the proxy collections (Fig. 6). With NTREND_T we find improvements everywhere, whereas B14_T shows more regions with negative than positive skill (note that we use PSM with only temperature). The PAGES data set has mainly positive skill, but negative skill in a large region

around the Himalaya and in some parts of North America. This suggest that using moisture sensitive proxies to reconstruct temperature as in B14_T, which works just because temperature and precipitation are correlated at a given location, is not ideal. Hence, further experiments with an improved PSM and upgraded screening procedure were conducted to take the proxies' temperature or moisture sensitivity better into account and to find an option to use the PAGES and B14 collection at locations, where no expert selected proxies are available but rather keep

the quality of the expert selected data, where it is available.

Before we come to a more sophisticated PSM and more sophisticated input data screening, we simply combine all three data sets using still a model with only temperature (ALL_T). This experiment performs well. Temperature correlation now reaches levels of the NTREND_T experiment, where NTREND data is available and additionally correlation improvements cover the regions, where only PAGES or B14 have data (Fig. 3d). RMSESS values are

positive in most regions, too. However, around India and the Himalaya negative skill is likely related to the impact of the PAGES data whereas negative skill in the US southwest seems the results from B14 data modeled as temperature only. Precipitation correlations improved only marginally (Fig. 4d) and precipitation RMSESS (Fig. 7d) is mostly negative.

The obvious change to improve precipitation reconstruction skill is to use a PSM that includes precipitation, i.e.

a multiple regression model with 12 coefficients for temperature and precipitation influence during the 6-months growing season (experiment ALL_TR). Temperature correlation and skill remains at the same high level (Fig. 3e and 6e), but precipitation correlations improve everywhere, particularly over North America (Fig. 4e). Precipitation RMSESS values become positive in most regions, too (Fig. 7e). The only exceptions are the Himalaya region and most northeast of Russia.

So far, we have not excluded any proxies from the data assimilation. We trust that proxies with no or a weak climate signal simply have regression coefficient close to zero and large residuals. This way they hardly affect the analysis. However, in a regression model with 12 independent variables and only 70 years of overlapping data, some records may just by chance get more weight than they deserve. Therefore, our next step is the introduction of a weak screening. In a first step, we only assimilate proxies with p-values < 0.05 for the full regression model

(ALL_TR_scr0.05). This removes ca. 16% of the proxies and hardly affects correlations (Fig. 3f, 4f, 5f) but removes most of the negative Asian *RMSESS* values in both, temperature and precipitation (Fig. 6f and 7f).

This result appears good, but this could also be a result of overfitting the regression model because any combination of growing season months was allowed to affect tree growth. It would not make physiological sense if a tree would be limited for instance by May, July and September temperatures but not by June and August

temperatures. Hence, the next step is to further constrain the model. The tree growth should be affected by climate conditions in a locally varying growing season of consecutive months. We fit all possible combinations of

temperature and precipitation influences in consecutive months and chose the model with the lowest AIC (note that additionally duplicates are removed; experiment ALL_TR_scr0.05_AIC_NOdup). As a result of this more physically based growth model, reconstruction skill decreases slightly in some regions with a high number of paleodata such as parts of China and parts of North America (Fig. 6g and 7g). Because we only identify a few duplicate records, this suggests that the previously noted improvement in *RMSESS* was indeed partly due to overfitting. Nevertheless, temperature and precipitation correlations remain on the same high level everywhere (Fig. 3g, 4g). Sea-level correlation changes are still small and negative in China and at the Westcoast of North America (Fig. 5g).

Recently, Valler et al. (2018) could show that major improvements of the method used in this study can be achieved by using a background error covariance matrix, which is not only calculated from the 30 ensemble members for the current year (Franke et al. 2017) but blended with a climatological error covariance matrix based on random years and ensemble members from the original model simulations (see methods, experiment ALL_TR_scr0.05_AIC_NOdup_ClimCovar). Using improved covariance information increases *RMSESS* values again and a much smaller number of grid boxes with negative skill remains. Moreover, the largest effects of the better error covariance estimation appear in variables that have not been assimilated such as sea level pressure (Fig. 5h). This is very important because one of the reasons for using data assimilation instead to traditional statistical reconstruction techniques is the possibility to gain knowledge about further variables in a physically consistent way, which allows for a better dynamic interpretation of the identified climatic variations.

**4 Discussion**

Correlations of the reconstructions with temperature improved as it would be expected after the assimilation of the three data sets and using a temperature PSM. We calculate the regression coefficients based on instrumental temperature. Hence, all proxies that correlate in some way with instrumental temperature will be used to update the analysis temperature. The analysis has highest correlations improvements with instrumental temperature if the proxies themselves had highest correlations, which is the case for the NTREND data set with the best temperature proxies only. Correlations improvements are lower but cover a larger area with the B14 collection.

Note that correlation improvements can be a result of a negative relationship between tree-ring width and instrumental temperature if local growth is moisture limited and growing season temperature and precipitation are negatively correlated. This can be a benefit because through the covariance we use the extra information that dry summers are also warm and vice versa. Hence, we find much better precipitation correlation with the B14 collection than with the NTREND data set. However, using moisture sensitive trees to update temperature fields may cause problems. Precipitation variability shows high inter-annual variability in many locations but neither the same inter- to multi-decadal variability as temperature nor its centennial trend (Hartmann et al., 2013; Landrum et al., 2013). Although not an issue addressed in this work, another study suggests that including the unscreened B14 records and modeling them using a similar approach than presented herein (including both temperature and moisture influences), can lead to problems in the representation of longer than inter-annual scales in temperature reconstructions (Tardif et al., 2019).

The regression model is calibrated on the interannual time scale assuming that TRW limitations are time-independent. However, this may not be the case (Babst et al. 2019), and therefore decadal-to-multidecadal

variability may be less well represented. A similar argument holds for the update introduced by the model covariance matrix, which, although state dependent, may yield optimal estimates only for seasonal and not decadal time scales. However, our approach avoids these pitfalls in two ways. First, at multidecadal and longer time-scales, the model takes over, and therefore relations in our reconstructions are not constrained to be stationary across time scales. Furthermore, with our approach, the stationarity assumption is restricted to the regression model, thus it is a local stationarity - no further stationarity assumption concerning spatial variability is introduced except for experiment ALL_TP_scr0.05_AIC_NOdup_ClimCovar, where 50% of the background error covariance matrix is climatological and thus stationary. Most other approaches assume stationary spatial covariances.

Theoretically, it would be optimal to assimilate all available data and let each record be weighted based by its error. However, the true observation error is unknown and its estimation is uncertain. In our case, we use a multiple regression proxy-system model with 6/12 variables (six months of temperature and optionally six months of precipitation) in a 70-year period of overlapping instrumental data and proxy measurements to estimate regression coefficients. This rather short period and large number of independent variables can lead to overfitting the model and thus underestimating the observation error, which is defined by the regression residuals. Together with the low signal-to-noise ratio of many tree-ring chronologies, this can lead to an over- or under-correction of the model field in the assimilation step. An additional experiment with doubled observation error (not shown) increases $RMSESS$ values clearly. This suggests that PSM overfitting and consequently too small regression residuals are part of the reason for the negative $RMSESS$ skill scores in the B14_T experiment in contrast to the NTREND_T experiment (Fig. 4a and c).

In the following experiments (ALL_TR_scr0.05, ALL_TR_scr0.05_AIC_NOdup, ALL_TR_scr0.05_AIC_NOdup_ClimCovar) we tried to reduce the consequences of uncertain error estimates step by step. Excluding proxies without a significant climate signal ($p<0.05$) for the full regression model, clearly improves the $RMSESS$ skill score for temperature and precipitation in large parts of Asia (Fig. 6f and 7f). This highlights the negative effects of spurious correlation – even if it is very weak – on the analysis. Hence, screening the data appears to be important, especially in data sparse regions, where there is no chance for better records with smaller errors to correct errors introduced due to spurious covariances. In other reconstruction methods, for instance principal component regression or the search for the best analogs, screening of records will additionally be necessary to avoid spatial biases due to non-homogeneous proxy distributions (Bradley, 1996; Rutherford et al., 2005). However, this is negligible in the data assimilation framework because the number of assimilated records has a regional instead of global impact and because the method provides a measure of uncertainty in form of ensemble spread at each grid cell.

In the experiment, in which we only allow for a single growing season (ALL_TR_scr0.05_AIC_NOdup) per year instead of a statistically optimal selection of months and by removing duplicate records that are in more than one of the data collections, correlations improve slightly but $RMSESS$ decreases slightly. Obviously, we continue with this more realistic setup, but note that the choice what is "best" depends on the chosen statistic or the reconstruction characteristics that are wished by the user. For instance, correlation just measures covariance whereas $RMSESS$ is based on squared errors and hence penalizes especially large biases, i.e. it favors an underestimation of variability over an overestimation.

Finally, we introduce an improved background error covariance estimation scheme (ALL_TR_scr0.05_AIC_NOdup_ClimCovar, Valler et al. 2019). Because assimilated information is spread in space and in between variables through the covariance matrix, it is important to estimate covariances well. Estimating covariance from both, the 30 members at the current time step and from climatology and then blending both information, especially improves our results for variables, which have not been assimilated such as sea level pressure (Fig. 5h).

In reality, climate signals in tree-ring proxies may be even more complicated than a function of moisture availability and growing season temperature. Limiting factors may change over time (Babst et al., 2019) or light availability may be important and not always be highly correlated with temperature, i.e. more diffuse light after volcanic eruptions may stimulate growth (Stine and Huybers, 2014). More sophisticated proxy system forward models such as VS-Lite (Tolwinski-Ward et al., 2011) could be used in data assimilation (Acevedo et al., 2016; Dee et al., 2016). In fact, we have applied VS-lite to all TRW records in B14 (Breitenmoser et al., 2014). Although these models are more realistic and represent for instance non-linear responses, they introduce new problems mainly related to model biases. This currently prevents them for being used more broadly (Dee et al., 2016).

Finally, we tested the order of assimilated data, because we assimilate data serially. In combination with using covariance localization, the order could influence the final reconstruction (Greybush et al., 2011). Assimilating the data from the best to worst record in terms of regression residuals and in opposite order from worst to best, hardly influenced correlation and *RMSESS* skill scores (not shown). Hence, we continue to assimilate records starting with the best ones, similar to traditional reanalysis, which sort observations from the largest to smallest expected variance reduction in the reanalysis (Slivinski et al., 2019; Whitaker et al., 2008).

Although our results are specifically valid only for the data assimilation method used herein, it is likely that methods with a similar structure, i.e. using PSMs and variations of Kalman filters, will have similar sensitivities to the selection of input data (Tardif et al., 2019). We expect that they are even valid for most climate field reconstruction techniques, because the basic principles of transferring proxy information to climatic variables and dealing with errors share common concepts across these methods. Even though all such methods include some routines to separate climatic information from non-climatic noise, in practice results can almost always be improved by pre-selecting the records with the highest information content, independent from the reconstruction technique applied (Neukom et al., 2019a; 2019b; Smerdon and Pollack, 2016 and references therein). This suggests that our results are qualitatively transferrable to climate field reconstruction methods in general.

**Conclusion**

In this study, we use existing proxy data collections to generate climate field reconstructions, as it is common practice. We are aware that this is not in all cases the main aim for which these data collections were compiled. Hence, we want to highlight the consequences of using the data set for field reconstructions. These results are not meant the rank any data set above another. Disadvantages of a data set in our setup are most probably a result of unintended usage.

How to choose input data for paleo data assimilation? We address this question by comparing three paleodata compilations of different sizes as well as using all data sets together in combination with various screening approaches.

Just using a large collection of proxy data (B14) does not lead to a skillful reconstruction. In contrast, just using a small expert selection of the best temperature proxies (NTREND) leads to a good high latitude temperature reconstruction but wastes the potential of modern data assimilation technique to reconstruct the 4-dimensional multi-variate state of the atmosphere. However, simply combing all available input data and leaving the weighting completely to a statistical model does not lead to optimal results, either. Rejecting records without a clear climatic signal, removing duplicates and using a physically plausible PSM altogether lead to a better reconstruction.

Hence the answer to our research question if it is better to assimilate all available proxy data or just the best expert selection has to be answered with: neither of the two is optimal. We achieve the best results in terms of correlation and *RMSESS*, if we use a large collection of proxy records. However, to make proper use of input data, which was not screened by experts, it is crucial to:

1. use proxy system models that properly represent the paleodata, here taking possible temperature and moisture limitations of tree growth into account.
2. use correct physical assumptions, in our case about tree growth, to avoid statistical overfitting.
3. remove input data with random, not significant climate signals.
4. care about overfitting (underestimation of errors)

For a future project, it would be very interesting to study how different reconstruction methods handle these three differently screened data sets to see, if these results are valid for other reconstructions methods, too?

**Author contribution**

JF had the initial idea for this paper and performed most of the analysis and drafted the manuscript. VV contributed to the code development. SB helped to shape the manuscript and experimental design. RN contributed additional analysis and all authors provided critical feedback and contributed to the writing of the manuscript.

**Competing interests**

There are no competing interests present.

**Acknowledgements**

This project was supported by the Swiss National Science Foundation project 162668 (RE-USE) and EU ERC project 787574 (PALAEO-RA). We like to thank CSCS for their support in conducting the ECHAM simulations.

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

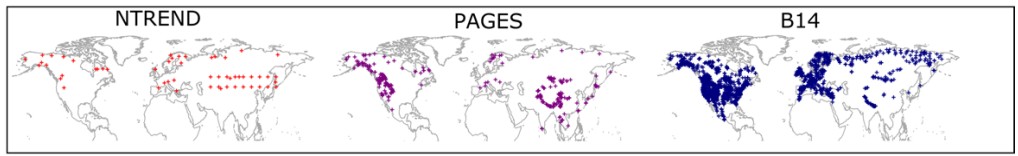

**Figure 1: Proxy locations of the three collections.**

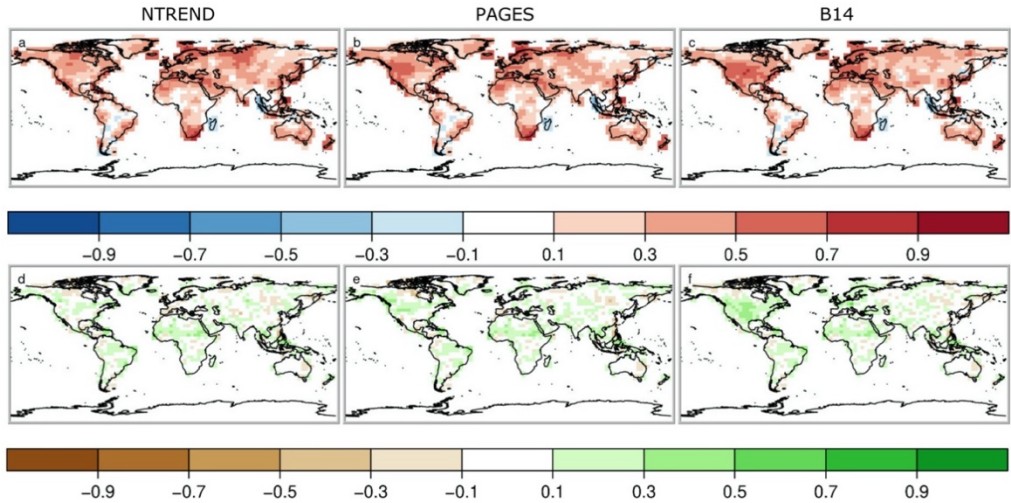

**Figure 2: Pearson correlations coefficients between the analysis and gridded instrumental data in the 20[th] century. The top panels show temperature and the bottom panels precipitation correlation. This figure shows results from experiments 1 to 3 (Table 1), i.e. after assimilation of the three proxy data collections using the proxy system model that assumes only growth limitation by temperature.**

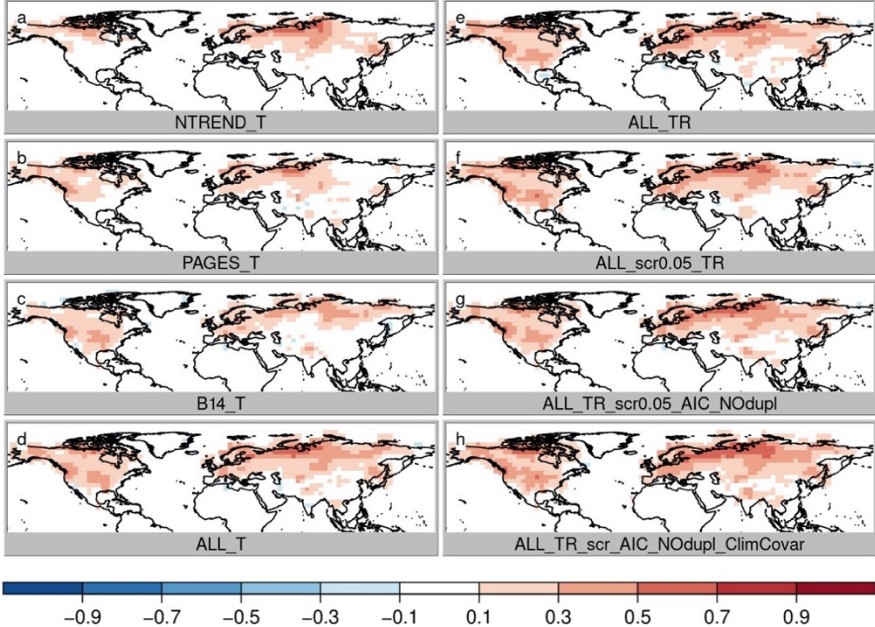

**Figure 3: Temperature correlation improvement of the analysis over the original model simulations, i.e. correlation between analysis and CRU TS minus correlation between simulations and CRU TS, where red colors indicate an improvement of the analysis. All maps show the Apr. to Sep. growing season of the northern hemisphere.**

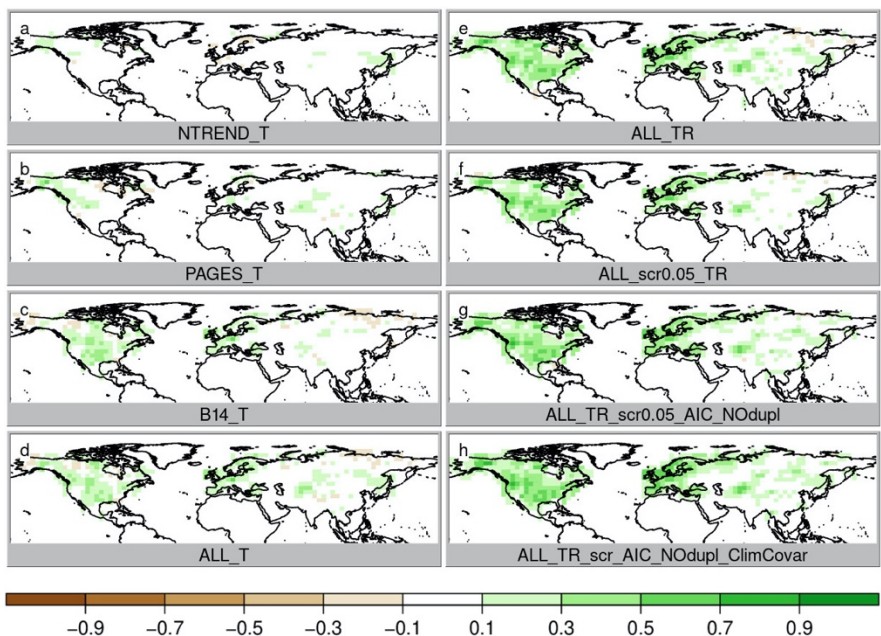

**Figure 4: Same as Fig. 3 for precipitation correlation, where green colors indicate an improvement of the analysis.**

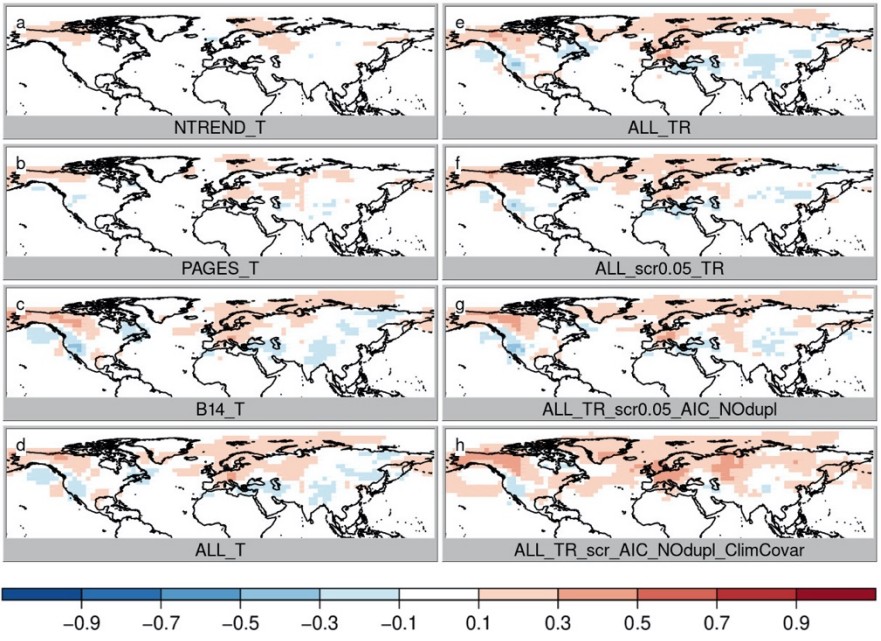

**Figure 5: Same as Fig. 3 for sea-level pressure correlation, where red colors indicate an improvement of the analysis.**

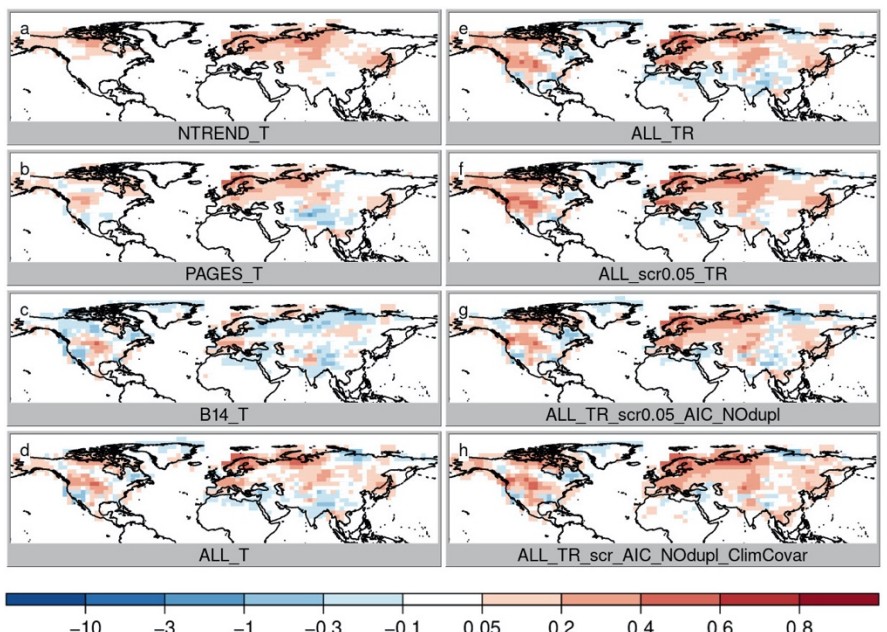

**Figure 6: Temperature RMSESS skill score, where red colors indicate an improvement of the analysis.**

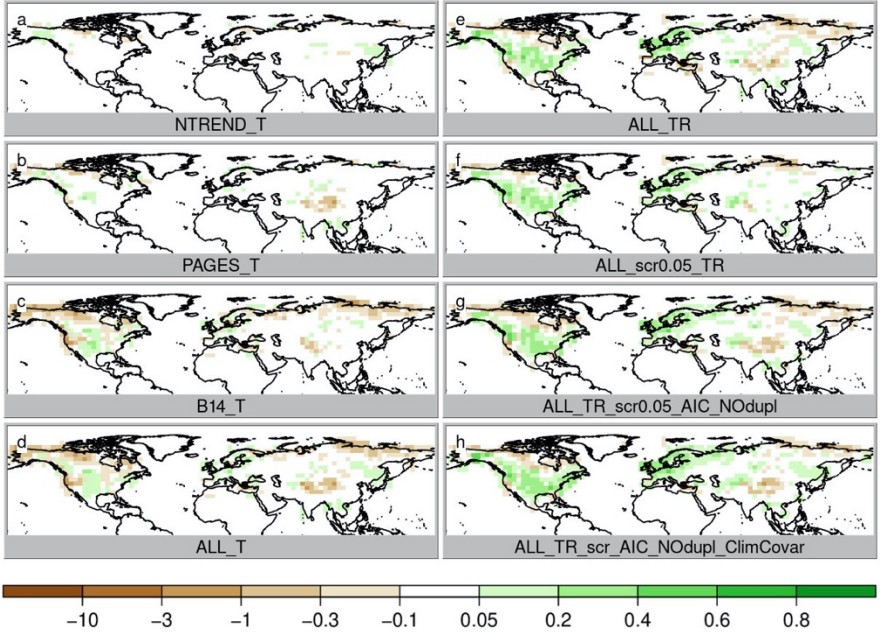

**Figure 7: Precipitation RMSESS skill score, where greens colors indicate an improvement of the analysis.**
