# Peer review of "The importance of input data quality and quantity in climate field reconstructions – results from the assimilation of various tree-ring collections"

_Climate of the Past, 2019_

## Referee Comment (RC1) · Anonymous Referee #1 · 11 Sep 2019

General comments:

The manuscript describes results from a series of data assimilation experiments aimed at identifying the "best" tree-ring input data set, i.e. the one leading to the largest improvements in paleoclimate reconstructions of temperature, sea-level pressure and precipitation. Three data sets are primarily tested, differing in the level of screening applied to tree-ring proxy records with respect to their climate sensitivity. The topic addressed in the manuscript is an important one and should be part of the published literature on climate field reconstructions. Several issues are found however, which

should be addressed before the manuscript is published.

Issues believed to be most important are:

1- The title of the manuscript is somewhat misleading. It implies that a more comprehensive evaluation is presented, covering a wider range of proxy archives, while the study is restricted to tree-ring data.

2- The presentation of results is problematic in several aspects:

2.1- The evaluation of the reconstructions is restricted to the instrumental-era, based on comparisons with the CRU data set for temperature and precipitation. This leads to several questions:

a. The validation is performed with the same data set used for the calibration of the forward models. What is the impact of this lack of independence on the overall conclusions of the study?

b. An evaluation of reconstructions limited to the instrumental era does not provide a solid perspective of variability over longer time scales. For instance, Tardif et al. (2019) have recently shown that the selection of assimilated tree-ring width data sets leads to noticeable differences in reconstructed temperature variability at multi-decadal to centennial time scales, including the representation of notable epochs such as the LIA. Is the long-term variability in your reconstructions affected by the use of various tree-ring data sets and how? Are your results consistent with dependencies to assimilated data shown by Tardif et al.?

2.2- The use of global maps in the presentation of the results is not optimal. The proxy data sets and related impact are confined to northern Hemisphere as is noted in the paper, with no signal elsewhere. Please use maps of NH only, which would show the results more clearly.

2.3- The results are shown from the perspective of changes in verification scores in the reconstructions over corresponding values from the prior. You should show and

discuss prior verification scores to cast your results in their proper context.

3- The impact of assimilated data is usually tied to the particular forward models (here proxy system models or PSMs) used. Yet, there is lack of information about the performance of the various proposed PSMs, nor a reference to prior work is given which would provide the necessary information. A characterization of the PSMs themselves would help the reader gain a more complete perspective on the results.

More specific comments/questions are:

- Page 1, line 19: the use of "best possible" seems an overstatement. Perhaps you mean the best reconstruction given the parameters tested?

- Page 1, line 23: how is "insignificant" defined in the present context? Please clarify.

- Page 2, line 37: The use of "probably" is not appropriate. There is a large body of literature on the impact of input data on data assimilation results (mostly focused on weather applications however). I believe a more unequivocal statement would better convey what is already known about the importance of the quantity and quality of input data to data assimilation systems.

- Page 2, line 41: Could you better explain/justify why the study is restricted to tree-ring data?

- Page 2, lines 48-49: Reference to specific studies which support your "would always be beneficial" statement would help improve the manuscript.

- Page 2, lines 52-53: The statement including "which often results in a small sample, uncertain residuals and possible model overfitting" lacks support. Can you include references or show results that highlights these problems?

- Page 2, line 75: Is your statement about "no dating uncertainties" accurate? Perhaps "small dating uncertainties" would be more appropriate?

- Page 3, line 85: About the statement "experts from various regional groups were

differently strict in their screening procedure", has this been characterized in a more formal way? Please provide support for this statement.

- Page 3, line 86: You mention that "N-TREND is a collection of 54 tree-ring reconstructions". Do you assimilate the reconstruction data or the tree-ring data underlying the reconstructions? Please clarify. If you use the reconstructions, please justify.

- Page 3, line 93: Statement with "...simulate tree-ring observations using modeled temperature or precipitation": I believe you also use PSMs that include both temperature and precipitation as input. A more accurate statement would therefore include "temperature and/or precipitation".

- Page 3, line 95: You use a single seasonal response for all records, and for temperature and precipitation. Please justify.

- Page 3, lines 97-103: I do not easily understand the information provided in this paragraph. I would suggest revising the description of the PSMs, perhaps using equations or illustrations, to provide a description the reader will more easily understand.

- Page 3, line 109: The procedure described here amounts to some screening of the data that is not evaluated nor discussed further here. Perhaps it should be.

- Page 3, line 112: Please specify what is the source of the 30 ensemble members. This is not clearly identified here.

- Page 3, line 115: What is the localization applied when precipitation is involved? Please specify.

- Page 4, line 120: I am failing to understand the justification for using anomalies about 71-yr mean values, or the prior model states? proxies? Please describe and justify in more detail so the reader can understand.

- Page 4, line 124: Can you support the statement that the method is "expected to provide consistent skill at all time scales"?

- Page 5, top row of table, rightmost frame: Can you provide some evidence to support your statement that records "Probably included some moisture or partly moisture sensitive" ones?

- Page 6, lines 162-163, statement that "B14 provides temperature information in places where temperature is correlated with precipitation": while most likely true, this statement seems incomplete. B14 also contains temperature sensitive records. One could argue that temperature improvements are mostly related to the assimilation of such records, more so than through the process you describe here. Can you support and quantify your statement?

- Page 6, line 183: regions (plural).

- Page 7, line 195, "lost": I believe you mean "lowest".

- Page 7, line 198: Can you provide a more complete reasoning as to why you believe overfitting is the (main?) reason for the behavior described in this paragraph?

- Page 7, lines 225-226, statement about problems on longer time scales: The experiments discussed in the manuscript are not evaluated on that particularly sensitive issue, an important shortcoming of the study in my opinion. The fact that results from your experiments can not provide a clear contribution toward characterizing or resolving this issue should be acknowledged.

- Page 9, lines 278-279: A more complete statement should include a reference to the work of Dee et al. (2016) to indicate that application of VS-lite has additional limitations related to model biases. (Dee, S., Steiger, N. J., Emile-Geay, J., and Hakim, Gregory J., 2016: The utility of proxy system modeling in estimating climate states over the Common Era, J. Adv. Model. Earth Sys., 8, 1164–1179, doi: 10.1002/2016MS000677)

- Page 9, line 288: data sets (plural)

- Including a figure showing the location of the proxy records from the various data sets would strengthen the presentation.

---

## Referee Comment (RC2) · Anonymous Referee #2 · 4 Nov 2019

Review of

**'The importance of input data quality and quantity in climate field reconstructions – results from a Kalman filter based paleodat assimilation method.'**

by J. Franke V. Valler, S. Brönnimann, R. Neukom, and F. Jaume Santero

**Recommendation: minor revisions**

This manuscript analyses the effect of the choice of input data in a Kalman filter based assimilation method for climate proxy data. Given the increasing use of data assimilation methods in paleoclimatology this is a practically relevant question. The climate reconstructions derived by assimilating tree ring width data for the period 1901 – 1970 AD are evaluated against gridded observations for near surface temperature, sea level pressure and precipitation. The best reconstruction skill is found when using input data that are a compromise between good spatial coverage and selecting only proxies that are well linked to local temperature and precipitation.

Overall this is a good paper that provides relevant information for the paleoclimate community, and in principle I support publication. However, several points should be explained better or should be modified. They are listed below.

**Specific comments**

1)
The period which is analysed should be stated in the abstract. At the moment the first time this information is given is in line 130. The abstract should also briefly mention the type of assimilation method.

2)
Line 21, 'improved' relative to what? This does become clear later in the text, but the abstract needs to make sense on its own.

3)
The paper links the results in several places to terms in the Kalman filter, namely to the proxy system model, the observation error covariance matrix and the background error covariance matrix. These are important comments, but readers who are not experts in data assimilation will probably not understand them, because the Kalman filter equation that is used is not given in the paper. I do appreciate that these details are given in previous publications, but with respect to its core elements a paper should be self-contained. Please add the equation to the method section and discuss there how the terms are calculated and how information is spread by the various terms from the proxy data to the different reconstructed meteorological variables. When presenting the results please refer back to this discussion where appropriate.

4)
Line 92, 'we need a forward model that simulates them in the model state vector'. This is not well phrased.

5)
There should be clear comments to what extent the findings can be transferred to other data assimilation methods used in paleoclimatology. It is likely that methods with a similar structure, i.e. using PSMs and variations of Kalman filters, will have similar sensitivities to the selection of input data, while others, for instance particle filters, may not.

6)
Line 57-59. The statement on the similarity between the method used in this paper and the method used in the last millennium project is misleading. The method used in the paper uses a 'transient offline method', in which the background state is time-dependent due to the signal of the external forcing. This aspect is actually highlighted by the authors in lines 120-124 and lines 231-232. In contrast, the last millennium project uses a 'stationary offline method' in which the background state does not depend on time. This crucial difference should be mentioned.

7)
Lines 121-122. The comments on low-frequency variability should include a discussion of the setup for the simulations that provide the background state. Why is sea surface temperature mentioned as a forcing? Are the simulations done with atmosphere-only GCMs? If so, which sea surface temperatures are used? These comments should also discuss the role of random, internal, low-frequency variability.

It should also be clarified that the validation measures are calculated from annual values and are thus dominated by inter-annual variability. This fact and the short evaluation period imply that an evaluation of low-frequency variability is not possible in this study.

8)
Line 125, it is not clear from which data the running mean is calculated and what 'model' refers to.

9)
Line 131, 'just at correlation itself' sounds strange.

10)
Lines 137/139/264, 'punishes' should not be used in scientific writing.

11)
Line 138-140, please include a more detailed justification of why the evaluation is based on ensemble means rather than on individual ensemble members followed by averaging of the skill scores. This should include explicit statements on the effect of the reduced variability in ensemble means on the RE; the current statement is unclear.

12)
Line 227, 'TRW limitations remain the same' is not clear.

13)
Line 254, it should be 'principal' not 'principle'

14)
The use of hyphens is inconsistent and often wrong. Adjectives that are constructed from two words should usually be hyphenated. Examples are 'temperature-sensitive', 'regression-based', 'time-dependent' (which is better than 'time-variant' used in line 113), 'low-frequency' (if used

as an adjective), 'inter-annual', 'multi-decadal, 'multi-variate' etc. In some case it is also correct to combine the two words, e.g. 'multivariate'

15)
Line 195, replace 'lost' with 'lowest'

16)
Lines 212-213, This is not a proper sentence.

17)
The paper

Matsikaris, A., Widmann, M. and Jungclaus, J., 2016. Influence of proxy data uncertainty on data assimilation for the past climate. *Climate of the Past*, *12*(7), pp.1555-1563.

addresses similar questions and should be included in the introduction and/or the discussion.

---

## Author Comment (AC1) · 2 Dec 2019

General comments:

The manuscript describes results from a series of data assimilation experiments aimed at identifying the "best" tree-ring input data set, i.e. the one leading to the largest improvements in paleoclimate reconstructions of temperature, sea-level pressure and precipitation. Three data sets are primarily tested, differing in the level of screening applied to tree-ring proxy records with respect to their climate sensitivity. The topic addressed in the manuscript is an important one and should be part of the published literature on climate field reconstructions. Several issues are found however, which should be addressed before the manuscript is published.

**We appreciate that the reviewer considers this topic to be of general importance.**

Issues believed to be most important are:

1- The title of the manuscript is somewhat misleading. It implies that a more comprehensive evaluation is presented, covering a wider range of proxy archives, while the study is restricted to tree-ring data.

**We believe that these results are relevant for various archive type. However, the reviewer is right that we only tested tree-ring proxies. Hence, we will choose a more precise title: "The importance of input data quality and quantity in climate field reconstructions – results from the assimilation of various tree-ring data collections"**

2- The presentation of results is problematic in several aspects:

2.1- The evaluation of the reconstructions is restricted to the instrumental-era, based on comparisons with the CRU data set for temperature and precipitation. This leads to several questions:

a. The validation is performed with the same data set used for the calibration of the forward models. What is the impact of this lack of independence on the overall conclusions of the study?

**The reviewer is right that there is a lack of independence which comes from 1) the regression model and 2) the residuals. Concerning 1), regression coefficients are estimated from gridded instrumental data sets to translate grid cell temperature (and moisture) anomalies to local tree-ring measurements. The optimization is done on tree rings, not on the climate data, and it is done on many local scales and not the large scale. In that sense the effects of the dependence are rather indirect. In contrast the**

**statistical reconstruction methods, which directly estimate a climate variable such as temperature through the regression parameter estimate, our assimilation method is less far affected by the calibration procedure. Nevertheless, we agree with the reviewer that using the same data for validation probably leads to a slight overestimation in reconstruction skill and this is the reason, why we made additional "leave-one-out"-experiments in the publication of the original reconstruction (Franke et al. 2017). Concerning 2), we use these regression residuals as an estimate of error covariance, i.e. the larger the residuals, the smaller the weight of the proxy observation in the assimilation process. Again, the error estimate concerns tree ring width, not climate parameters.**

**Note that in this study we just compare the relative skill of various inputs data sets, so the impact of dependencies will be the same for all. We do not see any reason how the relative skill should be influenced by not having a fully independent validation data.**

**In the revised manuscript, we will explain in the methods section, why this lack of independence cannot influence the finding of this study.**

b. An evaluation of reconstructions limited to the instrumental era does not provide a solid perspective of variability over longer time scales. For instance, Tardif et al. (2019) have recently shown that the selection of assimilated tree-ring width data sets leads to noticeable differences in reconstructed temperature variability at multi-decadal to centennial time scales, including the representation of notable epochs such as the LIA. Is the long-term variability in your reconstructions affected by the use of various tree- ring data sets and how? Are your results consistent with dependencies to assimilated data shown by Tardif et al.?

**Our results are complementary to the research of Tardif et al. (2019). They use a method in which the prior consists of an ensemble combined from random model years of the past millennium, i.e. the prior has no multi-decadal or centennial variability. In contrast, our method uses a transient ensemble simulation as a prior, i.e. the prior in each year is in agreement with the model forcings. We keep the multi-decadal to centennial variability of the model response to the forcings in our reconstruction and just assimilate 71-year running anomalies based on the fact that many to tree-ring chronologies do not contain the correct centennial variability (Franke et al 2011). Hence, we cannot draw any conclusions on the impact of the input data set choice in low-frequency variability. We touch upon this point in the second and third paragraph of the discussion but will make it clearer in the revised version. Already in the introduction, we will better explain the differences between the methods used by Tardif et al. and us.**

**Based on the comments of both reviewers, we will extend the entire methods section. This will allow the reader to better understand this study without reading the previous publications, which explained more details about the used method.**

2.2- The use of global maps in the presentation of the results is not optimal. The proxy data sets and related impact are confined to northern Hemisphere as is noted in the paper, with no signal elsewhere. Please use maps of NH only, which would show the results more clearly.

**Thanks for the suggestion, we will limit the maps to the northern hemisphere in the revised version.**

2.3- The results are shown from the perspective of changes in verification scores in the reconstructions over corresponding values from the prior. You should show and discuss prior verification scores to cast your results in their proper context.

**Our prior is a forced model simulation, which has by definition already some skill, e.g. positive correlation due to warming trend in the 20th century. Therefore, we present the improvements with regard to forced model simulations. This makes more sense as well with regard to comment 2.1.a. Nevertheless, we understand the point that it is of interest to the reader how well your prior already agrees with instrumental observations and how well the reconstruction in performs in comparison to instrumental data. We will add a figure with absolute skill of the prior and the reconstruction, highlighting that the method is able to generate a reasonable reconstruction.**

3- The impact of assimilated data is usually tied to the particular forward models (here proxy system models or PSMs) used. Yet, there is lack of information about the performance of the various proposed PSMs, nor a reference to prior work is given which would provide the necessary information. A characterization of the PSMs themselves would help the reader gain a more complete perspective on the results.

**The PSM is described in section 2, line 90. The annual tree-ring data are translated into temperature (and moisture) based on a multiple regression model using monthly means of a six months growing season (April to September in the northern hemisphere). Thus, there are 6 regression coefficients in the experiments with temperature only and 12 regression coefficients in the experiments where growth can be limited by temperature and precipitation. Because of a lack of more sophisticated PSMs for tree-ring density and possible model biases, which would affect non-linear growth functions, we decided to use a regression model instead of a more sophisticated tree-growth model such as VSL (Tolwinski-Ward et al. 2013).**

More specific comments/questions are:

- Page 1, line 19: the use of "best possible" seems an overstatement. Perhaps you mean the best reconstruction given the parameters tested?

**Will be stated more carefully.**

- Page 1, line 23: how is "insignificant" defined in the present context? Please clarify.

**We will clarify that the p-value of 0.05 of the regression-based proxy system model was used as a threshold.**

- Page 2, line 37: The use of "probably" is not appropriate. There is a large body of literature on the impact of input data on data assimilation results (mostly focused on weather applications however). I believe a more unequivocal statement would better convey what is already known about the importance of the quantity and quality of input data to data assimilation systems.

**We refer to the effect of input data quality in mostly statistical climate reconstructions, which is discussed to a much smaller degree in literature than in the case of data**

**assimilation for weather forecasting. However, the reviewer is right that paleo-climatologists could learn from research done in meteorology. We will add appropriate references to make this literature more known and remove the word "probably".**

- Page 2, line 41: Could you better explain/justify why the study is restricted to tree-ring data?

**We will explain that tree-ring proxies are the most widely used proxy types to reconstruct climate of the past centuries because they are the most abundant. Additionally, they are best suited for data-assimilation based reconstruction because they have hardly any dating uncertainties.**

- Page 2, lines 48-49: Reference to specific studies which support your "would always be beneficial" statement would help improve the manuscript.

**We will add a reference.**

- Page 2, lines 52-53: The statement including "which often results in a small sample, uncertain residuals and possible model overfitting" lacks support. Can you include references or show results that highlights these problems?

**Many tree-ring measurements were already done in the 1970th to 1990th. Hence, the number of proxy data drops rapidly from the 1970th to the present (e.g. J. Emile-Geay et al (2017). Instrumental station networks outside Europe and the United States of America, however, were very sparse before 1900 and only reach roughly present-day spatial coverage around 1950 (Jones et al., 2012, Fig. 1). Hence, many locations of temperature sensitive tree-ring proxies, which are located in remote mountainous regions or high latitudes neither have station measurements nearby nor long overlapping periods.**

- Page 2, line 75: Is your statement about "no dating uncertainties" accurate? Perhaps "small dating uncertainties" would be more appropriate?

**We will change it to "hardly any dating uncertainties".**

- Page 3, line 85: About the statement "experts from various regional groups were differently strict in their screening procedure", has this been characterized in a more formal way? Please provide support for this statement.

**The PAGES2k data set is a global proxy data collection gathered by multiple regional groups. While we use the entire collection, Emile-Geay et al. (2017) provide additionally multiple screening levels of the data (their Tab. 2). The amount of screened records varies by region if these stricter rules are applied (see supplementary Fig. S2, S3 and S4 in Emile-Geay et al., 2017).**

**The difference in the amount of data in the various regions is also caused by different priorities. The European group, for example, only included the longest and highest quality records from the wealth of datasets that exist in this region. In contrast, most other regional groups included all available records that fulfill the global minimum**

**selection criteria. Therefore, the number of records in the PAGES2k database from Asia and North America is much higher than from Europe.**

**We have amended the text to read: "This compilation represents a compromise of good quantity, large spatial coverage and good quality paleodata, based on global selection criteria. However, experts from various regional groups were differently strict in their screening procedure, which lead to varying data density in the different regions".**

- Page 3, line 86: You mention that "N-TREND is a collection of 54 tree-ring reconstructions". Do you assimilate the reconstruction data or the tree-ring data underlying the reconstructions? Please clarify. If you use the reconstructions, please justify.

**We will clarify in the text that we used the N-TREND tree-ring chronologies. In neither case we work with raw tree-ring measurement but use processed chronologies, in which multiple samples from one site have been combined, growth trends have been removed etc.**

- Page 3, line 93: Statement with "...simulate tree-ring observations using modeled temperature or precipitation": I believe you also use PSMs that include both temperature and precipitation as input. A more accurate statement would therefore include "temperature and/or precipitation".

**Yes, will be corrected accordingly.**

- Page 3, line 95: You use a single seasonal response for all records, and for temperature and precipitation. Please justify.

**No, we allow for all six months of a hemispheric growing season to potentially contain regression coefficients to be different from one. However, this also allows for having growths influenced by a few or a single month only. We will clarify this in the revised version.**

- Page 3, lines 97-103: I do not easily understand the information provided in this paragraph. I would suggest revising the description of the PSMs, perhaps using equations or illustrations, to provide a description the reader will more easily understand.

**We will rewrite this paragraph to clarify the PSMs (see last point).**

- Page 3, line 109: The procedure described here amounts to some screening of the data that is not evaluated nor discussed further here. Perhaps it should be.

**The same tree-ring site may have been included in multiple collections. To prevent the same observations from being assimilated multiple times, we assume that different study sites should be more than 0.1° apart from each other. Hence, this procedure is only a removal of duplicate records in the experiments, where data sets are combined. These rare cases hardly affect the reconstruction skill and hence there is no need for further discussion. We will clarify this in the revised text.**

- Page 3, line 112: Please specify what is the source of the 30 ensemble members. This is not clearly identified here.

**We add the references to the ECHAM simulation ensemble (CCC400) published in Bhend et al. (2012) and Franke et al. (2017).**

- Page 3, line 115: What is the localization applied when precipitation is involved? Please specify.

**We will add the equation used for localization including parameters for temperature and precipitation.**

- Page 4, line 120: I am failing to understand the justification for using anomalies about 71-yr mean values, or the prior model states? proxies? Please describe and justify in more detail so the reader can understand.

**The general problem in many tree-ring chronologies is the fact that they were not specifically created with the aim to retain realistic variability at all time scales. For instance, if a study aimed at interannual variations, multidecadal to centennial variability may have been filtered out. Or already the sampling strategy may not have been appropriate to retain such low frequency variability. Therefore, we only assume that tree-ring chronologies contain a reliable interannual to decadal signal. Accordingly, we assimilate anomalies around a 71-year mean. We will describe this procedure in more detail in the revised version of the manuscript.**

- Page 4, line 124: Can you support the statement that the method is "expected to provide consistent skill at all time scales"?

**Most tree-ring chronologies can be expected to represent interannual variability similarly well. However, centennial scale variability is not similarly well retained in all records, (see last comment or Franke et al. 2011). Therefore, we only assimilate anomalies around the 71-year mean and update the same 71-year anomaly field in the model. The model climatology is added again after the assimilation is finished. This way, the centennial-scale variability in our paleo-reanalysis is just a function of the model response to the external forcings and the model remains physically consistent but biased. We prefer this procedure not only because of proxy data characteristics but also because it does not introduce artificial biases when new observations become available (see Franke et al. 2017).**

- Page 5, top row of table, rightmost frame: Can you provide some evidence to support your statement that records "Probably included some moisture or partly moisture sensitive" ones?

**Many tree-ring records have a mixed climate signal, i.e. are not pure temperature or precipitation recorders. Many of the records in the PAGES2k database may have a significant precipitation signal, which may be even stronger that the temperature signal. Inclusion criteria were mainly that a record needs to be temperature sensitive, independent from potential relations to other variables, even if those are stronger. If you look at the proxy distribution maps in Emile-Geay et al. (2017), you can find many sites in warm and dry locations such as the south-western United States. These sites have been used in hydroclimatic reconstructions (Steiger et al. 2018, Cook et al, 2007).**

**In Emile-Geay et al. (2017) a sign correction is done, i.e. if temperature and precipitation are negatively correlated, tree-ring chronologies can remain as temperature sensitive in the data set, no matter if they show an anomalously wide or a narrow ring in an anomalously warm growing season.**

- Page 6, lines 162-163, statement that "B14 provides temperature information in places where temperature is correlated with precipitation": while most likely true, this statement seems incomplete. B14 also contains temperature sensitive records. One could argue that temperature improvements are mostly related to the assimilation of such records, more so than through the process you describe here. Can you support and quantify your statement?

**We will check the sign of the regression coefficients of the assimilated observations in this region to evaluate how many temperature-sensitive observations are assimilated there. These should show wide rings in warm and dry years.**

- Page 6, line 183: regions (plural).

**Will be corrected.**

- Page 7, line 195, "lost": I believe you mean "lowest".

**Exactly**

- Page 7, line 198: Can you provide a more complete reasoning as to why you believe overfitting is the (main?) reason for the behavior described in this paragraph?

**As mentioned in one of the earlier comments, our regression model always contains coefficients for the six months of the growing season although growth may in many cases be limited to the shorter period. In such cases, regression coefficients will be close to zero but will not be exactly zero because we work with a relatively small sample size. Hence, in many cases we use a regression model with too many independent variables, which do not contain information. In multiple regression, this is known to cause model overfitting.**

- Page 7, lines 225-226, statement about problems on longer time scales: The experiments discussed in the manuscript are not evaluated on that particularly sensitive issue, an important shortcoming of the study in my opinion. The fact that results from your experiments can not provide a clear contribution toward characterizing or resolving this issue should be acknowledged.

**As explained above, this study is complementary to Tardif et al. 2019 in this respect and we cannot draw any conclusion in this regard with our method because our low-frequency variability is the response of the model to the forcings and not proxy dependent. This will be clear, when we have already explained our method in more detail in the methods section.**

- Page 9, lines 278-279: A more complete statement should include a reference to the work of Dee et al. (2016) to indicate that application of VS-lite has additional limitations related to model biases. (Dee, S., Steiger, N. J., Emile-Geay, J., and Hakim, Gregory J., 2016: The

utility of proxy system modeling in estimating climate states over the Common Era, J. Adv. Model. Earth Sys., 8, 1164–1179, doi: 10.1002/2016MS000677)

**Will be added.**

- Page 9, line 288: data sets (plural)

**Will be corrected.**

- Including a figure showing the location of the proxy records from the various data sets would strengthen the presentation.

**Will be added.**

**References, which had not been included in the discussion paper:**

**P. D. Jones, D. H. Lister, T. J. Osborn, C. Harpham, M. Salmon, and C. P. Morice, "Hemispheric and large-scale land-surface air temperature variations: An extensive revision and an update to 2010," J. Geophys. Res., vol. 117, no. 5, p. D05127, Mar. 2012.**

**E. R. Cook, R. Seager, M. A. Cane, and D. W. Stahle, "North American drought: Reconstructions, causes, and consequences," Earth Science Reviews, vol. 81, no. 1, pp. 93–134, Mar. 2007.**

---

## Author Comment (AC2) · 2 Dec 2019

**Reviewer 2**

Review of '**The importance of input data quality and quantity in climate field reconstructions – results from a Kalman filter based paleodat assimilation method.**' by J. Franke V. Valler, S. Brönnimann, R. Neukom, and F. Jaume Santero

**Recommendation: minor revisions**

This manuscript analyses the effect of the choice of input data in a Kalman filter based assimilation method for climate proxy data. Given the increasing use of data assimilation methods in paleoclimatology this is a practically relevant question. The climate reconstructions derived by assimilating tree ring width data for the period 1901 – 1970 AD are evaluated against gridded observations for near surface temperature, sea level pressure and precipitation. The best reconstruction skill is found when using input data that are a compromise between good spatial coverage and selecting only proxies that are well linked to local temperature and precipitation.

Overall this is a good paper that provides relevant information for the paleoclimate community, and in principle I support publication. However, several points should be explained better or should be modified. They are listed below.

**We appreciate that the reviewer considers this study to be of interest to the paleoclimate community.**

**Specific comments**

1) The period which is analysed should be stated in the abstract. At the moment the first time this information is given is in line 130. The abstract should also briefly mention the type of assimilation method.

**This information on the method and time period will already be added to the abstract.**

2) Line 21, 'improved' relative to what? This does become clear later in the text, but the abstract needs to make sense on its own.

**Will be changed to "…but fail to provide information for other regions and other variables".**

3) The paper links the results in several places to terms in the Kalman filter, namely to the proxy system model, the observation error covariance matrix and the background error covariance matrix. These are important comments, but readers who are not experts in data assimilation will probably not understand them, because the Kalman filter equation that is used is not given in the paper. I do appreciate that these details are given in previous publications, but with respect to its core elements a paper should be self-contained. Please add the equation to the method section and discuss there how the terms are calculated and how information is spread by the various terms from the proxy data to the different reconstructed meteorological variables. When presenting the results please refer back to this discussion where appropriate.

**We will add more details about the method in the revised version to allow reader to understand this paper without reading previous publications about the method.**

4) Line 92, 'we need a forward model that simulates them in the model state vector'. This is not well phrased.

**Will be rephrased.**

5) There should be clear comments to what extent the findings can be transferred to other data assimilation methods used in paleoclimatology. It is likely that methods with a similar structure, i.e. using PSMs and variations of Kalman filters, will have similar sensitivities to the selection of input data, while others, for instance particle filters, may not.

**We think that the quality/selection of input data has similar consequences not only in other data assimilation methods but in statistical reconstruction, too. However, there will be method dependent differences. We will add a paragraph on this topic to the discussion.**

6) Line 57-59. The statement on the similarity between the method used in this paper and the method used in the last millennium project is misleading. The method used in the paper uses a 'transient offline method', in which the background state is time-dependent due to the signal of the external forcing. This aspect is actually highlighted by the authors in lines 120-124 and lines 231-232. In contrast, the last millennium project uses a 'stationary offline method' in which the background state does not depend on time. This crucial difference should be mentioned.

**We will add this information already here in the introduction to avoid confusion.**

7) Lines 121-122. The comments on low-frequency variability should include a discussion of the setup for the simulations that provide the background state. Why is sea surface temperature mentioned as a forcing? Are the simulations done with atmosphere-only GCMs? If so, which sea surface temperatures are used? These comments should also discuss the role of random, internal, low-frequency variability.

**As mentioned above, we will extend the methods section that it allows the reader to understand the entire study without reading previous publications on the method before.**

It should also be clarified that the validation measures are calculated from annual values and are thus dominated by inter-annual variability. This fact and the short evaluation period imply that an evaluation of low-frequency variability is not possible in this study.

**Yes, as explained in the answers to the first reviewer's comments, our methodology is not suitable to draw conclusions on the proxy data sets' influence on centennial-scale climate variability. This study is complementary to the results found by Tardif et al. (2019) on low frequency variability effects caused by the input data selection.**

8) Line 125, it is not clear from which data the running mean is calculated and what 'model' refers to.

**As mentioned before, we will explain the method in greater detail in the revised manuscript.**

9) Line 131, 'just at correlation itself' sounds strange.

**Will be corrected to "Instead of analyzing absolute correlation coefficients, we analyze correlation improvements …"**

10) Lines 137/139/264, 'punishes' should not be used in scientific writing.

**Will be replaced by "penalize".**

11) Line 138-140, please include a more detailed justification of why the evaluation is based on ensemble means rather than on individual ensemble members followed by averaging of the skill scores. This should include explicit statements on the effect of the reduced variability in ensemble means on the RE; the current statement is unclear.

**Thanks for this suggestion. You are right that we can expect reduced variability in the ensemble mean compared to the ensemble members and compared to the validation data, too. Nevertheless, most user will be interested in the ensemble mean and its skill. One would expect that the ensemble mean of the transient simulations before assimilation has little variability and that the assimilation would lead to an increase in variability bringing it closer to the observations. This is also what we observe with the generally positive RE skill scores. This way, a perfect skill score of one could only be reached, if the ensemble would have no variability left and perfectly matches the observations. We do not expect that the pattern or sign of the skill score would change in case of averaging the skill of the single members. However, the skill scores of the single members and hence also their mean could be expected to be larger than the skill score of the ensemble mean because variance is not systematically underestimated (see Fig. 6 in Bhend et al., 2012). We will test this and justify in the revised text why we present the results in one way or the other.**

12) Line 227, 'TRW limitations remain the same' is not clear.

**We will clarify this by writing: "TRW limitations are time-independent".**

13) Line 254, it should be 'principal' not 'principle'

**Will be corrected.**

14) The use of hyphens is inconsistent and often wrong. Adjectives that are constructed from two words should usually be hyphenated. Examples are 'temperature-sensitive', 'regression-based', 'time-dependent' (which is better than 'time-variant' used in line 113), 'low-frequency' (if used as an adjective), 'inter-annual', 'multi-decadal, 'multi-variate' etc. In some case it is also correct to combine the two words, e.g. 'multivariate'

**Thanks for the explanation, we will check the entire document.**

15) Line 195, replace 'lost' with 'lowest'

**Will be corrected.**

16) Lines 212-213, This is not a proper sentence.

**Will be rephrased.**

17) The paper Matsikaris, A., Widmann, M. and Jungclaus, J., 2016. Influence of proxy data uncertainty on data assimilation for the past climate. *Climate of the Past*, *12*(7), pp.1555-1563. addresses similar questions and should be included in the introduction and/or the discussion.

**We will add this reference and report/discuss their findings.**

---

## Referee Comment (RC3) · Edward Cook (Referee) · 6 Dec 2019

To begin, let me declare that I am not an expert on the new data assimilation methods (DA) being used for climate field reconstruction (CFR) now. Therefore, I will not comment on the way in which the "state-of-the-art paleo data assimilation approach" has been applied in this paper. Rather, I will stick more so to what the title of the paper indicates, i.e. "the importance of input data quality and quantity in climate field reconstructions" as a generic problem that spans all methods of CFR. In so doing, I will point out what I regard as a problem with one of the main conclusions of this paper.

[Figure]

This study is based on three collections of tree-ring records: "(1) 54 of the best temperature sensitive tree-ring chronologies chosen by experts; (2) 415 temperature sensitive tree-ring records chosen less strictly by regional working groups and statistical screening; (3) 2287 tree-ring series that are not screened for climate sensitivity." These are the N-TREND, PAGES2K, and B14 data sets, respectively. I will not get into the issue of how the tree-ring series were processed (detrended and standardized) for temperature reconstruction other than to say that it is crucial to the recovery of multi-decadal to centennial timescale variability. This is possible from tree rings as numerous published studies have shown, but it is a difficult problem nonetheless. Regarding this study, the processing methods used are likely to vary considerably between the three data sets used, with the 54 best N-TREND tree-ring chronologies processed best by the experts, but the effects of these differences are not possible to determine in this paper. This is not a criticism. It is just the way it is given the data used.

The importance of input data quality and quantity in climate field reconstructions is at a basic level a given, so much of what this paper demonstrates is not terribly surprising. Thus, as a first-order conclusion, data quality and quantity do matter and more of both is better than less. However, as the authors show, quantity does not necessarily help if the quality of climate signal in the tree rings is not also considered given the target variable being reconstructed, in this case temperature. Thus, data screening for the signal of interest can have a big impact on the quality of the climate field reconstructions produced. The generic process of data screening in dendroclimatology goes back many years of course (e.g., Fritts, 1962), so again there is no surprise here. What is more controversial is the use of precipitation-sensitive tree-ring series to reconstruct past temperature through an inverse evapotranspiration demand mediated temperature signal rather than through a direct temperature effect on tree growth. I will not dwell on this here because it appears to work okay in certain cases, e.g. Trouet et al. (2013). However, there remains some concern about how the power spectrum of temperature reconstructions based on these quite different tree growth signals may differ. Let's just say that an inverse temperature signal is not as optimal as the direct one used in Wilson

et al. (2016) and should be used with caution.

What has not been adequately considered in the paper, however, are differences in the size and location of the proxy domains used in the CFR experiments relative to the size and location of the climate field being reconstructed. For example, there is a great difference in size and location between the domain occupied by the 54 N-TREND series and the temperature domain being reconstructed. This basic issue was investigated by Kutzbach and Guetter (1980) in their classic paper on paleoenvironmental network design. It is not often cited today, yet should be mandatory reading for anyone who wishes to engage in CFR. In it, Kutzbach and Guetter (1980) show that reconstructing a large climate field from a much smaller proxy field is likely to be far less effective compared to the case where the proxy field is large and extends beyond the limits of the climate field being reconstructed. Such is clearly not the case regarding the N-TREND data used in this paper's CFR experiments.

The N-TREND data are exclusively from the 40°-75°N region rather than over the much larger domains of the other two tree-ring data sets. As such, those 54 tree-ring chronologies were never intended to be used in the way done in this paper because the temperature signals in many of the N-TREND series are comparatively local and therefore most reliable at that spatial scale of the overall N-TREND domain. See Anchukaitis et al. (2017) for Part 2 of the N-TREND study and the maps contained therein. Thus, the statement in the Abstract ''. . . nor the small expert selection [N-TREND] leads to the best possible climate field reconstruction" is really quite unfair because the experiments in this paper were set up in almost the worst possible way for N-TREND to succeed well. Thus, I find the results of this study difficult to interpret because of the vastly different spatial sampling that exists between N-TREND and the other two tree-ring datasets relative to the temperature field being reconstructed.

The authors also talk about assessments of reconstruction skill or skill improvement, but this is not really true in the classical sense where estimates are compared to actual data not used in the model calibration exercise. ÂăSo, there is no true out-of-sample

skill assessment made in their analyses and estimates of true reconstruction skill remain unknown. This is basically acknowledged by the authors in lines 127-128: "it must be noted that the final reconstruction is consistent only in the model world." Yet, true model validation tests could have been made by reserving a traditional validation interval for testing as is typically done in classical statistical CFR. This can be done in the context of data assimilation for CFR too as discussed in Steiger et al. (2018). The authors could, for example, calibrate the proxies only back to 1920 and check performance of the reconstructions over the withheld interval for skill and clues of overfitting. How ever done, some form of out-of-sample model validation testing should be mandatory when applying and testing any CFR method.

More specifically, a statistic called the root-mean-square-error skill score (RE) is used in this paper to compare the relative performances of the tree-ring data sets used in the DA experiments. But there is some unwanted and unnecessary confusion here. The 'true' RE (Reduction of Error) has a long history of use in both meteorology (Lorenz, 1956) and paleoclimatology (Fritts, 1976) as a measure of skill of 'out-of-sample' forecasts and hindcasts, respectively. To use the RE as classically defined requires an explicit calibration interval for model development and estimation of its mean state (climatology) and an explicit validation interval for testing the skill of the model estimates against withheld or 'out-of-sample' data. In this case, the minimum benchmark for model skill is RE > 0, i.e. skill > climatology. This does not appear to be the case here. Rather the authors seem to be using the model ensemble mean without proxy assimilation as the reference. ÂăAs such, there are not any explicitly defined calibration and validation intervals, and the authors are just assessing whether the simulations that assimilate the proxies do better than simulations that are merely forced with SSTs. Thus, the RE in this paper is very different from the classical RE of Lorenz (1956) and Fritts (1976) and should be called something else to avoid confusion.

Overall, I find this paper to be publishable after revisions that seriously consider the points raised in this review. However, I admit to not finding the results particularly

insightful either. They are pretty much as one would expect given the tree-ring data sets and experimental design used in this study.

References

Anchukaitis, K., Wilson, R., Briffa, K. Buentgen, U., Cook, E., D'Arrigo, R., Davi, N., Esper, J., Frank, D., Gunnarson, B., Hegerl, G., Helema, S., Klesse, S., Krusic, P., Linderholm, H.W., Myglan, V., Osborn, T., Rydval, M., Schneider, L., Schurer, A., Wiles, G., Zhang, P. and Zorita, E. 2017. Last millennium Northern Hemisphere summer temperatures from tree rings: Part II, spatially resolved reconstructions. Quaternary Science Reviews 163:1-22.

Fritts, H.C. 1962. An approach to dendroclimatology: screening by means of multiple regression techniques. Journal of Geophysical Research 67(4):1413-1420.

Fritts, H.C. 1976. Tree Rings and Climate. Academic Press, London.

Kutzbach, J.E. and Guetter, P.J. 1980. On the design of paleoenvironmental data networks for estimating large-scale patterns of climate. Quaternary Research 14:169-187.

Lorenz, E.N. 1956. Empirical orthogonal functions and statistical weather prediction. Scientific Report No. 1, Department of Meteorology, MIT, Cambridge, Mass.

Steiger, N.J., Smerdon, J.E., Cook, E.R. and Cook, B.I. 2018. A reconstruction of global hydroclimate and dynamical variables over the Common Era. Scientific Data 5:180086 doi:10.1086/sdata.2018.86.

Trouet, V., Diaz, H.F., Wahl, E.R., Viau, A.E., Graham, R., Graham, N., and Cook, E.R. 2013. A 1500-year reconstruction of annual mean temperature for temperate North America on decadal-to-multidecadal time scales. Environmental Research Letters 8:2-10.

---

## Author Comment (AC3) · 19 Dec 2019

To begin, let me declare that I am not an expert on the new data assimilation methods (DA) being used for climate field reconstruction (CFR) now. Therefore, I will not comment on the way in which the "state-of-the-art paleo data assimilation approach" has been applied in this paper. Rather, I will stick more so to what the title of the paper indicates, i.e. "the importance of input data quality and quantity in climate field reconstructions" as a generic problem that spans all methods of CFR. In so doing, I will point out what I regard as a problem with one of the main conclusions of this paper.

This study is based on three collections of tree-ring records: "(1) 54 of the best temperature sensitive tree-ring chronologies chosen by experts; (2) 415 temperature sensitive tree-ring records chosen less strictly by regional working groups and statistical screening; (3) 2287 tree-ring series that are not screened for climate sensitivity." These are the N-TREND, PAGES2K, and B14 data sets, respectively. I will not get into the issue of how the tree-ring series were processed (detrended and standardized) for temperature reconstruction other than to say that it is crucial to the recovery of multi-decadal to centennial timescale variability. This is possible from tree rings as numerous published studies have shown, but it is a difficult problem nonetheless. Regarding this study, the processing methods used are likely to vary considerably between the three data sets used, with the 54 best N-TREND tree-ring chronologies processed best by the experts, but the effects of these differences are not possible to determine in this paper. This is not a criticism. It is just the way it is given the data used.

Thank you for your feedback. We are aware of the spectral differences in proxies due to detrending, standardization, etc. and the first author of this paper even published on this topic (Franke et al., 2013). This inconsistency within the proxy data is actually the reason why we have another approach than most previous reconstructions, including the methodologically similar data assimilation approach used in the framework of the Last Millennium Reanalysis project (Hakim et al., 2016). A majority of reconstructions methods will probably be affected in their multi-decadal to centennial scale variability depending on the chosen input data set as shown by Tardif et al. (2019).

To avoid such issues and to be able to use the more reliable inter-annual to decadal variability that many tree-ring proxies contain even if they have not been specifically reconstructed to retain low frequency variability, we only assimilate anomalies around 71-year running means. As we explain in line 122, low frequency variability in our reconstruction is purely the model response to the external forcings and is consistent with model physics. As a consequence, the reviewer is right that we do not make use of the specific advantage of N-TREND to include probably the most realistic low frequency variability that can be obtained from tree-ring data.

The importance of input data quality and quantity in climate field reconstructions is at a basic level a given, so much of what this paper demonstrates is not terribly surprising. Thus, as a first-order conclusion, data quality and quantity do matter and more of both is better than less. However, as the authors show, quantity does not necessarily help if the quality of climate signal in the tree rings is not also considered given the target variable being reconstructed, in this case temperature. Thus, data screening for the signal of interest can have a big impact on the quality of the climate field reconstructions produced. The generic process of data screening in dendroclimatology goes back many years of course (e.g., Fritts, 1962), so again there is no surprise here. What is more controversial is the use of precipitation-sensitive tree-ring series to reconstruct past temperature through an inverse evapotranspiration demand mediated temperature signal rather than through a direct temperature effect on tree growth. I will not dwell on this here because it appears to work okay in certain cases, e.g. Trouet et al. (2013). However, there remains some concern about how the power spectrum of temperature reconstructions based on these quite different tree growth signals may differ. Let's just say that an inverse temperature signal is not as optimal as the direct one used in Wilson et al. (2016) and should be used with caution.

We completely agree that it is rather obvious and not new that more quantity and more quality is desirable. Nevertheless, there are constantly new collections of proxy data sets published and these are and will be used to

generate climate field reconstruction because the compilation a comprehensive proxy data set is a vast amount of work and requires experts from another field. All compilations have different strength and weaknesses and specific purposes, which they should be used for. Nevertheless, precipitation sensitive proxies remain in data sets that are specifically assembled for temperature reconstructions (Emile-Geay et al., 2017), which may in some instances be alright but not ideal as pointed out by the reviewer. Hence, proxy compilations are commonly used in other than just the best suited way, for instance because new methods allow to reconstruct multivariate 3-dimensional states of the atmosphere instead of surface temperature only. Hence, we find it useful to make users of these data sets aware of possible issues because not everyone involved in the development of reconstruction methods may be an expert in the proxy input data. Furthermore, it is not so obvious, how methods can deal with mixed temperature and precipitation signals in proxies and if the transfer of information into the multivariate atmospheric state works well. That is the reason why we include skill score maps for precipitation and sea level pressure, too.

What has not been adequately considered in the paper, however, are differences in the size and location of the proxy domains used in the CFR experiments relative to the size and location of the climate field being reconstructed. For example, there is a great difference in size and location between the domain occupied by the 54 N-TREND series and the temperature domain being reconstructed. This basic issue was investigated by Kutzbach and Guetter (1980) in their classic paper on paleoenvironmental network design. It is not often cited today, yet should be mandatory reading for anyone who wishes to engage in CFR. In it, Kutzbach and Guetter (1980) show that reconstructing a large climate field from a much smaller proxy field is likely to be far less effective compared to the case where the proxy field is large and extends beyond the limits of the climate field being reconstructed. Such is clearly not the case regarding the N-TREND data used in this paper's CFR experiments.

Thank you for suggesting this important study. Obviously, we also do not expect the N-TREND data set to produce a great reconstruction outside of the area covered although our method makes use of covariances in the model simulations between spatially distant locations. Our conclusions are not meant to criticize N-TREND for covering less space. Rather the opposite, we show that having this set of best reconstruction is greatly enhancing reconstruction skill in the covered areas and that these records get most weight in our assimilation procedure and therefore strongly influence the reconstruction. However, we wanted to highlight that a combination of data sets with less strictly selected records helps in regions where such high-quality information is not available. We find it noteworthy that at the same time data sets with less high-quality information do not blur the highest-quality information in regions covered by N-TREND.

The N-TREND data are exclusively from the 40°-75°N region rather than over the much larger domains of the other two tree-ring data sets. As such, those 54 tree-ring chronologies were never intended to be used in the way done in this paper because the temperature signals in many of the N-TREND series are comparatively local and therefore most reliable at that spatial scale of the overall N-TREND domain. See Anchukaitis et al. (2017) for Part 2 of the N-TREND study and the maps contained therein. Thus, the statement in the Abstract ''. . . nor the small expert selection [N-TREND] leads to the best possible climate field reconstruction" is really quite unfair because the experiments in this paper were set up in almost the worst possible way for N-TREND to succeed well. Thus, I find the results of this study difficult to interpret because of the vastly different spatial sampling that exists between N-TREND and the other two tree-ring datasets relative to the temperature field being reconstructed.

We will rephrase the statement in the abstract and also clearly this point in the discussion to avoid such a misunderstanding. However as explained above, we intend to show the consequences if proxy data compilations are used for climate field reconstruction in a way that was not really intended but occurs in practice. Nevertheless, even if not used in an ideal manner, N-TREND appears to be the most influential and important data set for all the region, which it covers. In this sense, our message is rather that such efforts as compiling the N-TREND are extremely important and helpful. However, if we aim at a global multivariate climate field reconstruction, we can add additional information if we combine data sets without blurring the information from high quality data sets.

The authors also talk about assessments of reconstruction skill or skill improvement, but this is not really true in the classical sense where estimates are compared to actual data not used in the model calibration exercise. So, there is no true out-of-sample skill assessment made in their analyses and estimates of true reconstruction skill remain unknown. This is basically acknowledged by the authors in lines 127-128: "it must be noted that the final reconstruction is consistent only in the model world." Yet, true model validation tests could have been made by reserving a traditional validation interval for testing as is typically done in classical statistical CFR. This can be done in the context of data assimilation for CFR too as discussed in Steiger et al. (2018). The authors could, for example, calibrate the proxies only back to 1920 and check performance of the reconstructions over the withheld interval for skill and clues of overfitting. However done, some form of out-of-sample model validation testing should be mandatory when applying and testing any CFR method.

We will add an additional figure, which shows the absolute skill of the reconstruction with respect instrumental data.

However, our methods is based on transient simulations as a prior in contrast to Steiger et al. (2018). Hence, our simulations already have skill and show for instance a greenhouse gas warming in the 20th century.

As we already explained to the first reviewer: "There is a lack of independence which comes from 1) the regression model and 2) the residuals. Concerning 1), regression coefficients are estimated from gridded instrumental data sets to translate grid cell temperature (and moisture) anomalies to local tree-ring measurements. The optimization is done on tree rings, not on the climate data, and it is done on many local scales and not the large scale. In that sense the effects of the dependence are rather indirect. In contrast the statistical reconstruction methods, which directly estimate a climate variable such as temperature through the regression parameter estimate, our assimilation method is less far affected by the calibration procedure. Nevertheless, we agree with the reviewer that using the same data for validation probably leads to a slight overestimation in reconstruction skill and this is the reason, why we made additional "leave-one-out"-experiments in the publication of the original reconstruction (Franke et al., 2017). Concerning 2), we use these regression residuals as an estimate of error covariance, i.e. the larger the residuals, the smaller the weight of the proxy observation in the assimilation process. Again, the error estimate concerns tree ring width, not climate parameters.

Note that in this study we just compare the relative skill of various inputs data sets, so the impact of dependencies will be the same for all. We do not see any reason how the relative skill should be influenced by not having a fully independent validation data.

In the revised manuscript, we will explain in the methods section, why this lack of independence cannot influence the finding of this study."

More specifically, a statistic called the root-mean-square-error skill score (RE) is used in this paper to compare the relative performances of the tree-ring data sets used in the DA experiments. But there is some unwanted and unnecessary confusion here. The 'true' RE (Reduction of Error) has a long history of use in both meteorology (Lorenz, 1956) and paleoclimatology (Fritts, 1976) as a measure of skill of 'out-of-sample' forecasts and hindcasts, respectively. To use the RE as classically defined requires an explicit calibration interval for model development and estimation of its mean state (climatology) and an explicit validation interval for testing the skill of the model estimates against withheld or 'out-of-sample' data. In this case, the minimum benchmark for model skill is RE > 0, i.e. skill > climatology. This does not appear to be the case here. Rather the authors seem to be using the model ensemble mean without proxy assimilation as the reference. As such, there are not any explicitly defined calibration and validation intervals, and the authors are just assessing whether the simulations that assimilate the proxies do better than simulations that are merely forced with SSTs. Thus, the RE in this paper is very different from the classical RE of Lorenz (1956) and Fritts (1976) and should be called something else to avoid confusion.

The idea of the "reduction of error" (RE) is to compare the error of a forecast to the error of a reference forecast. (Lorenz, 1956) used the root mean square error (RMSE) and climatology as a reference forecast. The same concept is also known as RMSE SS (Skill Score, Wilks, 2011) where the reference cannot only be climatology but also persistence or as in our case a transient model simulation forced not only by SSTs but solar variability, aerosols, land use changes, greenhouse gases, etc. and which already have RMSE SS > 0. Hence, achieving an RMSE SS > 0 in our case is much harder than just being better than climatology.

To avoid any confusion with the RE and CE definition used in the tree-ring community, we will call this skill score RMSE SS in the revised version.

**References**

Emile-Geay, J., McKay, N. P., Kaufman, D. S., Gunten, von, L., Wang, J., Anchukaitis, K. J., Abram, N. J., Addison, J. A., Curran, M. A. J., Evans, M. N., Henley, B. J., Hao, Z., Martrat, B., McGregor, H. V., Neukom, R., Pederson, G. T., Stenni, B., Thirumalai, K., Werner, J. P., Xu, C., Divine, D. V., Dixon, B. C., Gergis, J., Mundo, I. A., Nakatsuka, T., Phipps, S. J., Routson, C. C., Steig, E. J., Tierney, J. E., Tyler, J. J., Allen, K. J., Bertler, N. A. N., Björklund, J., Chase, B. M., Chen, M.-T., Cook, E., de Jong, R., DeLong, K. L., Dixon, D. A., Ekaykin, A. A., Ersek, V., Filipsson, H. L., Francus, P., Freund, M. B., Frezzotti, M., Gaire, N. P., Gajewski, K., Ge, Q., Goosse, H., Gornostaeva, A., Grosjean, M., Horiuchi, K., Hormes, A., Husum, K., Isaksson, E., Kandasamy, S., Kawamura, K., Kilbourne, K. H., Koç, N., Leduc, G., Linderholm, H. W., Lorrey, A. M., Mikhalenko, V., Mortyn, P. G., Motoyama, H., Moy, A. D., Mulvaney, R., Munz, P. M., Nash, D. J., Oerter, H., Opel, T., Orsi, A. J., Ovchinnikov, D. V., Porter, T. J., Roop, H. A., Saenger, C., Sano, M., Sauchyn, D., Saunders, K. M., Seidenkrantz, M.-S., Severi, M., Shao, X., Sicre, M.-A., Sigl, M., Sinclair, K., St George, S., St Jacques, J.-M., Thamban, M., Thapa, U. K., Thomas, E. R., Turney, C., Uemura, R., Viau, A. E., Vladimirova, D. O., Wahl, E. R., White, J. W. C., Yu, Z. and Zinke, J.: Data Descriptor: A global multiproxy database for temperature reconstructions of the Common Era, Sci. Data, 4, doi:10.1038/sdata.2017.88, 2017.

Franke, J., Brönnimann, S., Bhend, J. and Brugnara, Y.: A monthly global paleo-reanalysis of the atmosphere from 1600 to 2005 for studying past climatic variations, Sci. Data, 4, 170076, doi:10.1038/sdata.2017.76, 2017.

Franke, J., Frank, D., Raible, C. C., Esper, J. and Brönnimann, S.: Spectral biases in tree-ring climate proxies, Nature Climate change, 3(4), 360–364, doi:10.1038/nclimate1816, 2013.

Hakim, G. J., Emile-Geay, J., Steig, E. J., Noone, D., Anderson, D. M., Tardif, R., Steiger, N. and Perkins, W. A.: The last millennium climate reanalysis project: Framework and first results, J. Geophys. Res. Atmos., 121(1), 6745–6764, doi:10.1002/2016JD024751, 2016.

Lorenz, E. N.: Empirical Orthogonal Functions and Statistical Weather Prediction. 1956.

Steiger, N. J., Smerdon, J. E., Cook, E. R. and Cook, B. I.: A reconstruction of global hydroclimate and dynamical variables over the Common Era, Sci. Data, 5, 180086–15, doi:10.1038/sdata.2018.86, 2018.

Tardif, R., Hakim, G. J., Perkins, W. A., Horlick, K. A., Erb, M. P., Emile-Geay, J., Anderson, D. M., Steig, E. J. and Noone, D.: Last Millennium Reanalysis with an expanded proxy database and seasonal proxy modeling, Climate of the Past, 15(4), 1251–1273, doi:10.5194/cp-15-1251-2019, 2019.

Wilks, D. S.: Statistical Methods in the Atmospheric Sciences, Academic Press. 2011.

---

## Author Response (AR1)

[revised manuscript text omitted]

**Reviewer 1**
1- The title of the manuscript is somewhat misleading. It implies that a more comprehensive evaluation is presented, covering a wider range of proxy archives, while the study is restricted to tree-ring data.

**We changed the title to: "The importance of input data quality and quantity in climate field reconstructions – results from the assimilation of various tree-ring collections"**

2- The presentation of results is problematic in several aspects:

2.1- The evaluation of the reconstructions is restricted to the instrumental-era, based on comparisons with the CRU data set for temperature and precipitation. This leads to several questions:

a. The validation is performed with the same data set used for the calibration of the forward models. What is the impact of this lack of independence on the overall conclusions of the study?

**We added a discussion how this apparent lack of independence could affect our reconstruction and why it cannot have any influence on the conclusions drawn in the study (line 168ff).**

b. An evaluation of reconstructions limited to the instrumental era does not provide a solid perspective of variability over longer time scales. For instance, Tardif et al. (2019) have recently shown that the selection of assimilated tree-ring width data sets leads to noticeable differences in reconstructed temperature variability at multi-decadal to centennial time scales, including the representation of notable epochs such as the LIA. Is the long-term variability in your reconstructions affected by the use of various tree- ring data sets and how? Are your results consistent with dependencies to assimilated data shown by Tardif et al.?

**Our results are complementary to the research of Tardif et al. (2019). They use a method in which the prior consists of an ensemble combined from random model years of the past millennium, i.e. the prior has no multi-decadal or centennial variability. In contrast, our method uses a transient ensemble simulation as a prior, i.e. the prior in each year is in agreement with the model forcings. We keep the multi-decadal to centennial variability of the model response to the forcings in our reconstruction.**

**We explain this in detail in our new methods section and refer to this point in the discussion section, too.**

2.2- The use of global maps in the presentation of the results is not optimal. The proxy data sets and related impact are confined to northern Hemisphere as is noted in the paper, with no signal elsewhere. Please use maps of NH only, which would show the results more clearly.

**Thanks for the suggestion, we limited the maps to the northern hemisphere in the revised version.**

2.3- The results are shown from the perspective of changes in verification scores in the reconstructions over corresponding values from the prior. You should show and discuss prior verification scores to cast your results in their proper context.

**The new Fig. 2 shows correlation between our reconstructions and gridded instrumental data.**

3- The impact of assimilated data is usually tied to the particular forward models (here proxy system models or PSMs) used. Yet, there is lack of information about the performance of the various proposed PSMs, nor a reference to prior work is given which would provide the necessary information. A characterization of the PSMs themselves would help the reader gain a more complete perspective on the results.

**The PSM is now described in more detail in line 151ff.**

More specific comments/questions are:

- Page 1, line 19: the use of "best possible" seems an overstatement. Perhaps you mean the best reconstruction given the parameters tested?

**We rephrased this paragraph.**

- Page 1, line 23: how is "insignificant" defined in the present context? Please clarify.

**We clarified that the p-value of 0.05 of the regression-based proxy system model was used as a threshold.**

- Page 2, line 37: The use of "probably" is not appropriate. There is a large body of literature on the impact of input data on data assimilation results (mostly focused on weather applications however). I believe a more unequivocal statement would better convey what is already known about the importance of the quantity and quality of input data to data assimilation systems.

**We refer to the effect of input data quality in mostly statistical climate reconstructions, which is discussed to a much smaller degree in literature than in the case of data assimilation for weather forecasting. However, the reviewer is right that paleo-climatologists could learn from research done in meteorology. Therefore we added new references and removed the word "probably".**

- Page 2, line 41: Could you better explain/justify why the study is restricted to tree-ring data?

**We explain this now in lines 44f.**

- Page 2, lines 48-49: Reference to specific studies which support your "would always be beneficial" statement would help improve the manuscript.

**We added a reference.**

- Page 2, lines 52-53: The statement including "which often results in a small sample, uncertain residuals and possible model overfitting" lacks support. Can you include references or show results that highlights these problems?

**Many tree-ring measurements were already done in the 1970th to 1990th. Hence, the number of proxy data drops rapidly from the 1970th to the present (e.g. J. Emile-Geay et al (2017). Instrumental station networks outside Europe and the United States of America, however, were very sparse before 1900 and only reach roughly present-day spatial coverage around 1950. We added a reference for clarification.**

- Page 2, line 75: Is your statement about "no dating uncertainties" accurate? Perhaps "small dating uncertainties" would be more appropriate?

**We changed it to "hardly any dating uncertainties".**

- Page 3, line 85: About the statement "experts from various regional groups were differently strict in their screening procedure", has this been characterized in a more formal way? Please provide support for this statement.

**The PAGES2k data set is a global proxy data collection gathered by multiple regional groups. While we use the entire collection, Emile-Geay et al. (2017) provide additionally multiple screening levels of the data (their Tab. 2). The amount of screened records varies by region if these stricter rules are applied (see supplementary Fig. S2, S3 and S4 in Emile-Geay et al., 2017).**

**The difference in the amount of data in the various regions is also caused by different priorities. The European group, for example, only included the longest and highest quality records from the wealth of datasets that exist in this region. In contrast, most other regional groups included all available records that fulfill the global minimum selection criteria. Therefore, the number of records in the PAGES2k database from Asia and North America is much higher than from Europe.**

**We have amended the text to read: "This compilation represents a compromise of good quantity, large spatial coverage and good quality paleodata, based on global selection criteria. However, experts from various regional groups were differently strict in their screening procedure, which lead to varying data density in the different regions".**

- Page 3, line 86: You mention that "N-TREND is a collection of 54 tree-ring reconstructions". Do you assimilate the reconstruction data or the tree-ring data underlying the reconstructions? Please clarify. If you use the reconstructions, please justify.

**We clarified in the text that we used the N-TREND tree-ring chronologies. In neither case we work with raw tree-ring measurement but use processed chronologies, in which**

**multiple samples from one site have been combined, growth trends have been removed etc.**

- Page 3, line 93: Statement with "…simulate tree-ring observations using modeled temperature or precipitation": I believe you also use PSMs that include both temperature and precipitation as input. A more accurate statement would therefore include "temperature and/or precipitation".

**Yes, this has been corrected accordingly.**

- Page 3, line 95: You use a single seasonal response for all records, and for temperature and precipitation. Please justify.

**No, we allow for all six months of a hemispheric growing season to potentially contain regression coefficients to be different from one. However, this also allows for having growths influenced by a few or a single month only. We explain this now in detail in the new methods section.**

- Page 3, lines 97-103: I do not easily understand the information provided in this paragraph. I would suggest revising the description of the PSMs, perhaps using equations or illustrations, to provide a description the reader will more easily understand.

**This has been clarified in the new methods section.**

- Page 3, line 109: The procedure described here amounts to some screening of the data that is not evaluated nor discussed further here. Perhaps it should be.

**This in now explained in the methods section.**

- Page 3, line 112: Please specify what is the source of the 30 ensemble members. This is not clearly identified here.

**We added a short description of the simulations and references to the more detailed description of the simulations.**

- Page 3, line 115: What is the localization applied when precipitation is involved? Please specify.

**We added the equation used for localization including parameters for temperature and precipitation.**

- Page 4, line 120: I am failing to understand the justification for using anomalies about 71-yr mean values, or the prior model states? proxies? Please describe and justify in more detail so the reader can understand.

**The general problem in many tree-ring chronologies is the fact that they were not specifically created with the aim to retain realistic variability at all time scales. For instance, if a study aimed at interannual variations, multidecadal to centennial variability may have been filtered out. Or already the sampling strategy may not have been appropriate to retain such low frequency variability. Therefore, we only assume**

**that tree-ring chronologies contain a reliable interannual to decadal signal. Accordingly, we assimilate anomalies around a 71-year mean. We described this procedure in more detail in the methods sections of the new manuscript version.**

- Page 4, line 124: Can you support the statement that the method is "expected to provide consistent skill at all time scales"?

**Most tree-ring chronologies can be expected to represent interannual variability similarly well. However, centennial scale variability is not similarly well retained in all records, (see last comment or Franke et al. 2011). Therefore, we only assimilate anomalies around the 71-year mean and update the same 71-year anomaly field in the model. The model climatology is added again after the assimilation is finished. This way, the centennial-scale variability in our paleo-reanalysis is just a function of the model response to the external forcings and the model remains physically consistent but biased. We prefer this procedure not only because of proxy data characteristics but also because it does not introduce artificial biases when new observations become available (see Franke et al. 2017). This is now explained in the methods section, too.**

- Page 5, top row of table, rightmost frame: Can you provide some evidence to support your statement that records "Probably included some moisture or partly moisture sensitive" ones?

**Many tree-ring records have a mixed climate signal, i.e. are not pure temperature or precipitation recorders. Many of the records in the PAGES2k database may have a significant precipitation signal, which may be even stronger that the temperature signal. Inclusion criteria were mainly that a record needs to be temperature sensitive, independent from potential relations to other variables, even if those are stronger. If you look at the proxy distribution maps in Emile-Geay et al. (2017), you can find many sites in warm and dry locations such as the south-western United States. These sites have been used in hydroclimatic reconstructions (Steiger et al. 2018, Cook et al, 2007). In Emile-Geay et al. (2017) a sign correction is done, i.e. if temperature and precipitation are negatively correlated, tree-ring chronologies can remain as temperature sensitive in the data set, no matter if they show an anomalously wide or a narrow ring in an anomalously warm growing season.**

**This information has been included in Table 1.**

- Page 6, lines 162-163, statement that "B14 provides temperature information in places where temperature is correlated with precipitation": while most likely true, this statement seems incomplete. B14 also contains temperature sensitive records. One could argue that temperature improvements are mostly related to the assimilation of such records, more so than through the process you describe here. Can you support and quantify your statement?

**We checked the sign of the regression coefficients and found mostly negative relationships in the United States of America and the Mediterranean. Here, narrow rings indicate dry and warm growing seasons. This information has been included in the results.**

- Page 6, line 183: regions (plural).

**This has been corrected.**

- Page 7, line 195, "lost": I believe you mean "lowest".

**This has been corrected.**

- Page 7, line 198: Can you provide a more complete reasoning as to why you believe overfitting is the (main?) reason for the behavior described in this paragraph?

**As mentioned in one of the earlier comments, our regression model always contains coefficients for the six months of the growing season although growth may in many cases be limited to the shorter period. In such cases, regression coefficients will be close to zero but will not be exactly zero because we work with a relatively small sample size. Hence, in many cases we use a regression model with too many independent variables, which do not contain information. In multiple regression, this is known to cause model overfitting.**

**This is now explained in more detail in lines 256ff.**

- Page 7, lines 225-226, statement about problems on longer time scales: The experiments discussed in the manuscript are not evaluated on that particularly sensitive issue, an important shortcoming of the study in my opinion. The fact that results from your experiments can not provide a clear contribution toward characterizing or resolving this issue should be acknowledged.

**As explained above, this study is complementary to Tardif et al. 2019 in this respect and we cannot draw any conclusion in this regard with our method because our low-frequency variability is the response of the model to the forcings and not proxy dependent.**

**This should be clear after reading the new methods section.**

- Page 9, lines 278-279: A more complete statement should include a reference to the work of Dee et al. (2016) to indicate that application of VS-lite has additional limitations related to model biases. (Dee, S., Steiger, N. J., Emile-Geay, J., and Hakim, Gregory J., 2016: The utility of proxy system modeling in estimating climate states over the Common Era, J. Adv. Model. Earth Sys., 8, 1164–1179, doi: 10.1002/2016MS000677)

**This has been added in line 345.**

- Page 9, line 288: data sets (plural)

**This has been corrected.**

- Including a figure showing the location of the proxy records from the various data sets would strengthen the presentation.

**We added a new Fig. 1 with the proxy locations.**

**Reviewer 2**

Review of '**The importance of input data quality and quantity in climate field reconstructions – results from a Kalman filter based paleodata assimilation method.**' by J. Franke V. Valler, S. Brönnimann, R. Neukom, and F. Jaume Santero

**Specific comments**

1) The period which is analysed should be stated in the abstract. At the moment the first time this information is given is in line 130. The abstract should also briefly mention the type of assimilation method.

**The method and analyzed time period are now mentioned in the abstract.**

2) Line 21, 'improved' relative to what? This does become clear later in the text, but the abstract needs to make sense on its own.

**Has been changed to "…but fail to provide information for other regions and other variables".**

3) The paper links the results in several places to terms in the Kalman filter, namely to the proxy system model, the observation error covariance matrix and the background error covariance matrix. These are important comments, but readers who are not experts in data assimilation will probably not understand them, because the Kalman filter equation that is used is not given in the paper. I do appreciate that these details are given in previous publications, but with respect to its core elements a paper should be self-contained. Please add the equation to the method section and discuss there how the terms are calculated and how information is spread by the various terms from the proxy data to the different reconstructed meteorological variables. When presenting the results please refer back to this discussion where appropriate.

**We added a detailed methods section, which allows reader to understand this paper without reading previous publications about the method.**

4) Line 92, 'we need a forward model that simulates them in the model state vector'. This is not well phrased.

**This has been rephrased.**

5) There should be clear comments to what extent the findings can be transferred to other data assimilation methods used in paleoclimatology. It is likely that methods with a similar structure, i.e. using PSMs and variations of Kalman filters, will have similar sensitivities to the selection of input data, while others, for instance particle filters, may not.

**We think that the quality/selection of input data has similar consequences not only in other data assimilation methods but in statistical reconstruction, too. However, there will be method dependent differences.**

**We added a paragraph at the end of the discussion (line 354ff).**

6) Line 57-59. The statement on the similarity between the method used in this paper and the method used in the last millennium project is misleading. The method used in the paper uses a 'transient offline method', in which the background state is time-dependent due to the signal of the external forcing. This aspect is actually highlighted by the authors in lines 120-124 and lines 231-232. In contrast, the last millennium project uses a 'stationary offline method' in which the background state does not depend on time. This crucial difference should be mentioned.

**We already added this information to the introduction to avoid confusion (line 63ff).**

7) Lines 121-122. The comments on low-frequency variability should include a discussion of the setup for the simulations that provide the background state. Why is sea surface temperature mentioned as a forcing? Are the simulations done with atmosphere-only GCMs? If so, which sea surface temperatures are used? These comments should also discuss the role of random, internal, low-frequency variability.

**As mentioned above, we added a much more detailed methods section to the revised manuscript.**

It should also be clarified that the validation measures are calculated from annual values and are thus dominated by inter-annual variability. This fact and the short evaluation period imply that an evaluation of low-frequency variability is not possible in this study.

**Yes, as explained in the answers to the first reviewer's comments, we explain in the revised version that our methodology is not suitable to draw conclusions on the proxy data sets' influence on centennial-scale climate variability. This study is complementary to the results found by Tardif et al. (2019) on low frequency variability effects caused by the input data selection.**

8) Line 125, it is not clear from which data the running mean is calculated and what 'model' refers to.

**As mentioned before, everything is now explained in detail in the new method section.**

9) Line 131, 'just at correlation itself' sounds strange.

**This has been corrected to "Instead of analyzing absolute correlation coefficients, we analyze correlation improvements …"**

10) Lines 137/139/264, 'punishes' should not be used in scientific writing.

**This has been replaced by "penalize".**

11) Line 138-140, please include a more detailed justification of why the evaluation is based on ensemble means rather than on individual ensemble members followed by averaging of the skill scores. This should include explicit statements on the effect of the reduced variability in ensemble means on the RE; the current statement is unclear.

**Thanks for this suggestion. You are right that we can expect reduced variability in the ensemble mean compared to the ensemble members and compared to the validation**

**data, too. Nevertheless, most user will be interested in the ensemble mean and its skill. One would expect that the ensemble mean of the transient simulations before assimilation has little variability and that the assimilation would lead to an increase in variability bringing it closer to the observations. This is also what we observe with the generally positive RE skill scores. This way, a perfect skill score of one could only be reached, if the ensemble would have no variability left and perfectly matches the observations. We do not expect that the pattern or sign of the skill score would change in case of averaging the skill of the single members. However, the skill scores of the single members tend to be larger than the skill score of the ensemble mean because variance is not systematically underestimated.**

**We discuss this now in lines 199ff.**

12) Line 227, 'TRW limitations remain the same' is not clear.

**This has been changed to: "TRW limitations are time-independent".**

13) Line 254, it should be 'principal' not 'principle'

**This has been corrected.**

14) The use of hyphens is inconsistent and often wrong. Adjectives that are constructed from two words should usually be hyphenated. Examples are 'temperature-sensitive', 'regression-based', 'time-dependent' (which is better than 'time-variant' used in line 113), 'low-frequency' (if used as an adjective), 'inter-annual', 'multi-decadal, 'multi-variate' etc. In some case it is also correct to combine the two words, e.g. 'multivariate'

**We checked the document and hopefully corrected it everywhere.**

15) Line 195, replace 'lost' with 'lowest'

**This has been corrected.**

16) Lines 212-213, This is not a proper sentence.

**This sentence has been rephrased.**

17) The paper Matsikaris, A., Widmann, M. and Jungclaus, J., 2016. Influence of proxy data uncertainty on data assimilation for the past climate. *Climate of the Past*, *12*(7), pp.1555-1563. addresses similar questions and should be included in the introduction and/or the discussion.

**We now refer to this paper in line 53.**

**Reply to comments by Edward Cook**

To begin, let me declare that I am not an expert on the new data assimilation methods (DA) being used for climate field reconstruction (CFR) now. Therefore, I will not comment on the way in which the "state-of-the-art paleo data assimilation approach" has been applied in this paper. Rather, I will stick more so to what the title of the paper indicates, i.e. "the importance of input data quality and quantity in climate field reconstructions" as a generic problem that spans all methods of CFR. In so doing, I will point out what I regard as a problem with one of the main conclusions of this paper.

This study is based on three collections of tree-ring records: "(1) 54 of the best temperature sensitive tree-ring chronologies chosen by experts; (2) 415 temperature sensitive tree-ring records chosen less strictly by regional working groups and statistical screening; (3) 2287 tree-ring series that are not screened for climate sensitivity." These are the N-TREND, PAGES2K, and B14 data sets, respectively. I will not get into the issue of how the tree-ring series were processed (detrended and standardized) for temperature reconstruction other than to say that it is crucial to the recovery of multi-decadal to centennial timescale variability. This is possible from tree rings as numerous published studies have shown, but it is a difficult problem nonetheless. Regarding this study, the processing methods used are likely to vary considerably between the three data sets used, with the 54 best N-TREND tree-ring chronologies processed best by the experts, but the effects of these differences are not possible to determine in this paper. This is not a criticism. It is just the way it is given the data used.

**Thank you for your feedback. We are aware of the spectral differences in proxies due to detrending, standardization, etc. and the first author of this paper even published on this topic (Franke et al. 2011). This inconsistency within the proxy data is actually the reason why we have another approach than most previous reconstructions. To avoid such issues and to be able to use the more reliable inter-annual to decadal variability that many tree-ring proxies contain even if they have not been specifically reconstructed to retain low frequency variability, we only assimilate anomalies around 71-year running means. As we explain in line 146 of the revised manuscript, low frequency variability in our reconstruction is purely the model response to the external forcings and is consistent with model physics. As a consequence, the reviewer is right that we do not make use of the specific advantage of N-TREND to include probably the most realistic low frequency variability that can be obtained from tree-ring data.**

The importance of input data quality and quantity in climate field reconstructions is at a basic level a given, so much of what this paper demonstrates is not terribly surprising. Thus, as a first-order conclusion, data quality and quantity do matter and more of both is better than

less. However, as the authors show, quantity does not necessarily help if the quality of climate signal in the tree rings is not also considered given the target variable being reconstructed, in this case temperature. Thus, data screening for the signal of interest can have a big impact on the quality of the climate field reconstructions produced. The generic process of data screening in dendroclimatology goes back many years of course (e.g., Fritts, 1962), so again there is no surprise here. What is more controversial is the use of precipitation-sensitive tree-ring series to reconstruct past temperature through an inverse evapotranspiration demand mediated temperature signal rather than through a direct temperature effect on tree growth. I will not dwell on this here because it appears to work okay in certain cases, e.g. Trouet et al. (2013). However, there remains some concern about how the power spectrum of temperature reconstructions based on these quite different tree growth signals may differ. Let's just say that an inverse temperature signal is not as optimal as the direct one used in Wilson et al. (2016) and should be used with caution.

**We completely agree that it is rather obvious and not new that more quantity and more quality is desirable. Nevertheless, there are constantly new collections of proxy data sets published and these are and will be used to generate climate field reconstruction because the compilation a comprehensive proxy data set is a vast amount of work and requires experts from another field. All compilations have different strength and weaknesses and specific purposes, which they should be used for. Nevertheless, precipitation sensitive proxies remain in data sets that are specifically assembled for temperature reconstructions such as the PAGES data base. This may in some instances be alright but not ideal as pointed out by the reviewer. Hence, proxy compilations are commonly used in other than just the best suited way, for instance because new methods allow to reconstruct multivariate 3-dimensional states of the atmosphere instead of surface temperature only. Hence, we find it useful to make users of these data sets aware of possible issues because not everyone involved in the development of reconstruction methods may be an expert in the proxy input data. Furthermore, it is not so obvious, how methods can deal with mixed temperature and precipitation signals in proxies and if the transfer of information into the multivariate atmospheric state works well. That is the reason why we include skill score maps for precipitation and sea level pressure, too.**

What has not been adequately considered in the paper, however, are differences in the size and location of the proxy domains used in the CFR experiments relative to the size and location of the climate field being reconstructed. For example, there is a great difference in size and location between the domain occupied by the 54 N-TREND series and the temperature domain being reconstructed. This basic issue was investigated by Kutzbach and Guetter (1980) in their classic paper on paleoenvironmental network design. It is not often cited today, yet should be mandatory reading for anyone who wishes to engage in CFR. In it, Kutzbach and Guetter (1980) show that reconstructing a large climate field from a much smaller proxy field is likely to be far less effective compared to the case where the proxy field is large and extends beyond the limits of the climate field being reconstructed. Such is clearly not the case regarding the N-TREND data used in this paper's CFR experiments.

**Thank you for suggesting this important study. Obviously, we also do not expect the N-TREND data set to produce a great reconstruction outside of the area covered although our method makes use of covariances between spatially distant locations. Our conclusions are not meant to criticize N-TREND for covering less space. Rather the opposite, we show that having this set of best reconstruction is greatly enhancing**

**reconstruction skill in the covered areas and that these records get most weight in our assimilation procedure and therefore strongly influence the reconstruction. However, we wanted to highlight that a combination of data sets with less strictly selected records helps in regions where such high-quality information is not available. We find it noteworthy that at the same time data sets with less high-quality information do not blur the highest-quality information in regions covered by N-TREND.**

**We cite the study of Kutzbach and Guetter (1980) now and also explain that we cannot expect skill from the NTREND data set far outside the region that is covers.**

The N-TREND data are exclusively from the 40°-75°N region rather than over the much larger domains of the other two tree-ring data sets. As such, those 54 tree-ring chronologies were never intended to be used in the way done in this paper because the temperature signals in many of the N-TREND series are comparatively local and therefore most reliable at that spatial scale of the overall N-TREND domain. See Anchukaitis et al. (2017) for Part 2 of the N-TREND study and the maps contained therein. Thus, the statement in the Abstract ''. . . nor the small expert selection [N-TREND] leads to the best possible climate field reconstruction" is really quite unfair because the experiments in this paper were set up in almost the worst possible way for N-TREND to succeed well. Thus, I find the results of this study difficult to interpret because of the vastly different spatial sampling that exists between N-TREND and the other two tree-ring datasets relative to the temperature field being reconstructed.

**We rephrased the abstract to avoid such a misunderstanding. However as explained above, we intend to show the consequences if proxy data compilations are used for climate field reconstruction in a way that was not really intended but occurs in practice. Nevertheless, even if not used in an ideal manner, N-TREND appears to be the most influential and important data set for all the region, which it covers. In this sense, our message is rather that such efforts as compiling the N-TREND are extremely important and helpful. However, if we aim at a global multivariate climate field reconstruction, we can add additional information if we combine data sets without blurring the information from high quality data sets. This message should become clearer in the revised text.**

The authors also talk about assessments of reconstruction skill or skill improvement, but this is not really true in the classical sense where estimates are compared to actual data not used in the model calibration exercise. So, there is no true out-of-sample skill assessment made in their analyses and estimates of true reconstruction skill remain unknown. This is basically acknowledged by the authors in lines 127-128: "it must be noted that the final reconstruction is consistent only in the model world." Yet, true model validation tests could have been made by reserving a traditional validation interval for testing as is typically done in classical statistical CFR. This can be done in the context of data assimilation for CFR too as discussed in Steiger et al. (2018). The authors could, for example, calibrate the proxies only back to 1920 and check performance of the reconstructions over the withheld interval for skill and clues of overfitting. However done, some form of out-of-sample model validation testing should be mandatory when applying and testing any CFR method.

**We added a new Fig. 2, which shows the absolute skill of the reconstruction with respect to gridded instrumental data.**

**As we already explained to the first reviewer: "We added a discussion how this apparent lack of independence could affect our reconstruction and why it cannot have any influence on the conclusions drawn in the study (line 168ff).**

More specifically, a statistic called the root-mean-square-error skill score (RE) is used in this paper to compare the relative performances of the tree-ring data sets used in the DA experiments. But there is some unwanted and unnecessary confusion here. The 'true' RE (Reduction of Error) has a long history of use in both meteorology (Lorenz, 1956) and paleoclimatology (Fritts, 1976) as a measure of skill of 'out-of-sample' forecasts and hindcasts, respectively. To use the RE as classically defined requires an explicit calibration interval for model development and estimation of its mean state (climatology) and an explicit validation interval for testing the skill of the model estimates against withheld or 'out-of-sample' data. In this case, the minimum benchmark for model skill is RE > 0, i.e. skill > climatology. This does not appear to be the case here. Rather the authors seem to be using the model ensemble mean without proxy assimilation as the reference. As such, there are not any explicitly defined calibration and validation intervals, and the authors are just assessing whether the simulations that assimilate the proxies do better than simulations that are merely forced with SSTs. Thus, the RE in this paper is very different from the classical RE of Lorenz (1956) and Fritts (1976) and should be called something else to avoid confusion.

**The idea of the "reduction of error" (RE) is to compare an error of a forecast for instance the root mean square error (RMSE) to the error of a reference forecast. Lorenz 1956 used climatology as a reference forecast. The same concept is also known as RMSE SS (Skill Score, Wilks, 2011) where the reference cannot only be climatology but also persistence or as in our case a transient model simulation. Hence, achieving an RMSE SS > 0 in our case is much harder than just being better than climatology.**

**To avoid any confusion with the RE and CE definition used in the tree-ring community, we called this skill score RMSE SS in the revised version.**

---

## Author Response (AR2)

**Answers to reviewer comments on cp-2019-80: The importance of input data quality and quantity in climate field reconstructions – results from the assimilation of various tree-ring collections**

The revised manuscript is much improved. I suggest relatively minor, but important, modifications to the text.

I find the revised manuscript significantly improved over the original submission. I appreciate the efforts by the authors in addressing the issues raised by the reviewers. The description of the methods and presentation of the results are found to be clearer and more insightful. Some remaining minor issues are found in the revision, outlined below.

*We would like to thank the reviewer for the helpful comments and addressed all points mentioned below.*

Page 1, first line of abstract: Two very broadly defined factors in the quality of reconstructions are listed, methods and input data. Are there other factors? The use of "mainly" in the sentence seems to imply that other factors are at play. Perhaps delete the word?

*The word "mainly" has been deleted.*

Page 2, line 40: "the switch from radiosonde to satellite observations", I believe a more accurate statement would say "the addition of satellite to radiosonde observations".

*Has been corrected accordingly.*

Page 2, lines 52-53: I believe that your statement about some studies claiming that "higher quantity of input data would always be beneficial" is somewhat of a mischaracterization. Some of these studies do show that input of more data can have detrimental consequences on some of the characteristics of the reconstructions. Please revise.

*Has been revised to: A number of previous studies based on data assimilation techniques tended to assimilate a high quantity of input data instead of applying a strict data selection beforehand.*

Page 3 line 109: "…fact the our prior", "that" instead of "the"?

*Has been corrected.*

Page 4, second line: "only one observation per year per record" to be more precise?

*Has been corrected accordingly.*

Page 5, line 181: "little too much weight" is overly qualitative. This can still have some detrimental consequences and a statement reflecting that should be included in my opinion.

*We added: Uncertainty estimation in both, observations and models, is a crucial but challenging part of data assimilation. We evaluate the spread-to-error ratios to assess the under/overconfidence of our reconstructions.*

Page 6, Table 1, line 4: "duplicate proxies cannot be excluded": I believe they can, but in this experiment they are not. I would propose "duplicate proxies are not excluded" for a more accurate statement.

*Has been corrected accordingly.*

Table 1, line 6, "got little weight": is this clearly shown later or an hypothesis? As you explain later, relying on statistics can lead to mischaracterized error characteristics and weighting in the analysis. Please revise.

*We explain now that this basic statistical screening is the next step in the sensitivity experiments that removes records without or with very little and uncertain climatic information.*

Page 7, lines 220-223: If the majority of B14 records in the US do not contain information on temperature, and are appropriately weighted by the assimilation, why do we see an improved correlation, particularly over the

eastern US, when B14 is used over PAGES records? If included but with little weight in the analysis, I would expect a neutral effect on skill over the prior. How do you reconcile? Please explain more clearly.

Explanations can be found in the second paragraph of the discussion section: "Note that correlation improvements can be a result of a negative relationship between tree-ring width and instrumental temperature if local growth is moisture limited and growing season temperature and precipitation are negatively correlated. This can be a benefit because through the covariance we use the extra information that dry summers are also warm and vice versa. Hence, we find much better precipitation correlation with the B14 collection than with the NTREND data set."

Page 8, line 229 "Here, we find large differences": Please indicate a corresponding figure to guide the reader.

Has been added.

Page 8, line 231, "is again in the middle": This is not clear. Please provide a more quantitative description of the comparison.

Has been added.

Page 8, line 240, "correlation improvements": Please indicate a corresponding figure to guide the reader.

Has been added.

Page 8, your statement on lines 240-241: Just trying to summarize here: these regions of negative skill seem to be related to a combination of negative impact of PAGES data (Himalaya) (interestingly!), and B14 data forward modeled as temperature data only (India and US southwest), where N_TREND data do not influence positively. I would suggest that such details are worth including in the text.

Has been added.

Page 8, line 242: Fig. 4d instead of Fig. 6d ?

Has been corrected.

Page 8, lines 246-247, "precipitation skill clearly improves": Looking closely at your results, I would add that this improvement is particularly noticeable over North America, even perhaps most over the eastern US. This is a finding consistent with results presented in Tardif et al 2019. The authors could consider mentioning this fact to further underline their finding.

Has been corrected accordingly.

Page 8, sentence on lines 247-248: Are you referring to only precipitation here, or temperature as well. Please clarify.

Has been added.

Page 8, line 255, "removes most of the negative RMSESS": I feel this is a bit of an overstatement with respect to temperature. Negative RMSESS values persist over eastern North America for instance. The improvement seems more pronounced over Asia.

Has been specified: "removes most of the negative Asian RMSESS"

Page 8, line 264, "due to overfitting": And maybe due to duplicates? How can you distinguish? Please clarify.

We now explain: "Because we only identify a few duplicate records, this suggests …"

Page 9, line 272, "very few grid points with negative skill": I again feel like this is a bit of an overstatement. I do not see changes in precipitation skill, some improvement in temperature, but negative RMSESS values remain over parts of North America (eastern Canada, southern US).

We rephrased this: "and a much smaller number of grid boxes"

Page 9, line 273: I find it awkward that results shown in Fig. 5 are not discussed before this point, and that only one frame of the figure is discussed in the entire paper.

In the previous version, we only described sea-level pressure for the last experiment because the reconstruction skill was relatively low in the other experiments. Now, we describe sea-level in the results of other experiments as well.

Page 9, lines 291-292: This is not a very clear statement. I would suggest that a more accurate one would read something like: Although not an issue addressed in this work, another study suggests that including the unscreened B14 records and modeling them using a similar approach than presented herein (including both temperature and moisture influences), can lead to problems in the representation of longer (than inter-annual) scales in temperature reconstructions.

Has been changed accordingly.

Page 10, lines 312-313: I am not sure I understand how increasing observation errors and its associated decrease in analysis skill can be interpreted as a sign of overfitting. Can you describe more clearly the link?

We extended the explanation: "This suggests that PSM overfitting and consequently too small regression residuals are part of the reason for the negative RMSESS skill scores …"

Page 10, line 333: "underestimation of variability" ?

Has been corrected.

Page 11, lines 355-356: I believe that Tardif et al. (2019) have shown a sensitivity as well. Maybe cite their work to support your statement.

Citation has been added.

Page 11, line 367: "most probably"?

Has been corrected.

[revised manuscript text omitted]

**Commented [JF1]:** Reviewer comment: If the majority of B14 records in the US do not contain information on temperature, and are appropriately weighted by the assimilation, why do we see an improved correlation, particularly over the eastern US, when B14 is used over PAGES records? If included but with little weight in the analysis, I would expect a neutral effect on skill over the prior. How do you reconcile? Please explain more clearly.

Answer: Explanations can be found in the second paragraph of the discussion section: "
[revised manuscript text omitted]

600